# Prebiotic gas flow environment enables isothermal nucleic acid replication

**Philipp Schwintek, Emre Eren, Christof Bernhard Mast, Dieter Braun***

Systems Biophysics, Physics department, Center for NanoScience, Ludwig-Maximilians-Universität München, Munich, Germany

**\*For correspondence:**
dieter.braun@lmu.de

**Competing interest:** The authors declare that no competing interests exist.

## eLife Assessment

This **important** work shows how a simple geophysical setting of gas flow over a narrow channel of water can create a physical environment that leads to the isothermal replication of nucleic acids. The work presents **compelling** evidence for an isothermal polymerase chain reaction in careful experiments involving evaporation and convective flows, complimented with fluid dynamics simulations. This work will be of interest to scientists working on the origin of life and more broadly, on nucleic acids and diagnostic applications.

**Abstract** Nucleic acid replication is a central process at the origin of life. On early Earth, replication is challenged by the dilution of molecular building blocks and the difficulty of separating daughter from parent strands, a necessity for exponential replication. While thermal gradient systems have been shown to address these problems, elevated temperatures lead to degradation. Also, compared to constant temperature environments, such systems are rare. The isothermal system studied here models an abundant geological environment of the prebiotic Earth, in which water is continuously evaporated at the point of contact with the gas flows, inducing up-concentration and circular flow patterns at the gas-water interface through momentum transfer. We show experimentally that this setting drives a 30-fold accumulation of nucleic acids and their periodic separation by a threefold reduction in salt and product concentration. Fluid dynamic simulations agree with observations from tracking fluorescent beads. In this isothermal system, we were able to drive exponential DNA replication with Taq polymerase. The results provide a model for a ubiquitous non-equilibrium system to host early Darwinian molecular evolution at constant temperature.

## Introduction

The emergence of life on Earth is still an unsolved puzzle to contemporary research. It is estimated that this event dates back approximately 3.7–4.5 billion years, with fossil carbon isotope signatures being the oldest evidence for life around 3.7 billion years ago (*Pearce et al., 2018*; *Rosing, 1999*). In order to reconstruct how early molecular life began before this time, it is crucial to identify and understand plausible geological environments, which support early prebiotic reaction networks that could have led to the life we know today (*Orgel, 1994*).

The common theory is that the Darwinian evolution of informational polymers was at the core of the origin of life (*Orgel, 1994*). Among these, nucleic acids, like RNA, stand out for their capability to both store genetic information and catalyze their own replication through transient formation of double-stranded helices (*Gilbert, 1986*). These abilities allow them to mutate and evolve, enabling them to adapt to diverse environments and eventually encode, build, and utilize proteins as the catalysts used in modern life.

Dilution, however, poses a significant obstacle, since such prebiotic reactions require sufficiently high concentrations of their reagents to work (*Luisi, 2015*). Large reservoirs, such as the ocean, cannot compensate for diffusion, because they lack local sources of energy to drive reaction pathways out of equilibrium (*Goldenfeld and Woese, 2011*). The resulting homogeneity renders these environments unlikely to have harbored early molecular life (*Lane et al., 2010*).

Local physical non-equilibria, however, have shown the ability to up-concentrate molecules, such as nucleic acids, in a variety of different geological settings (*Ianeselli et al., 2023*). Examples range from thermal gradients in rock pores, local evaporation, re-hydration cycles of warm ponds, adsorption to mineral surfaces, heated gas bubbles in porous rocks, foams, or the eutectic phase in freeze-thaw cycles (*Mast et al., 2013*; *Matreux et al., 2024*; *Pearce et al., 2017*; *Damer and Deamer, 2015*; *Ferris et al., 1996*; *Morasch et al., 2019*; *Tekin et al., 2022*; *Trinks et al., 2005*; *Stribling and Miller, 1991*).

However, the accumulation of salts and molecules comes at a cost. Single-stranded nucleic acids replicate into double-stranded forms. These strands must separate again to complete a full replication cycle. But strand separation becomes increasingly difficult after accumulation, because the melting temperature of oligonucleotides is strongly dependent on the local salt concentration (*Schildkraut and Lifson, 1965*). Despite high $Mg^{2+}$ concentrations being required for replication and catalytic activity (*Hampel and Cowan, 1997*), they can elevate the melting temperature of nucleic acid duplex structures to levels surpassing even the boiling point of water (*Szostak, 2012*). Oligonucleotides readily hydrolyze into nucleotide fragments under these conditions, rendering high temperature spikes as a primary strand separation mechanism more detrimental than beneficial (*Li and Breaker, 1999*).

Therefore, other mechanisms are required at the origin of life to separate nucleic acid strands with minimal thermal stress, and at best combined with an environment where supplied biomolecules are accumulated from the environment and trapped for long periods of time. Examples have used pH oscillations to drive nucleic acid strand separation, which can be caused either by differential thermophoresis of ionic species or by periodic freeze-thaw cycles (*Mariani et al., 2018*; *Keil et al., 2017*; *Takenaka et al., 2006*). Also, dew droplet cycles in a rock pore subjected to a temperature gradient can periodically melt strands by transiently lowering the salt concentration (*Ianeselli et al., 2019*; *Salditt et al., 2023*). Heated gas-water interfaces were also shown to promote many prebiotic synthesis reactions (*Deal et al., 2021*; *Morasch et al., 2019*; *Dass et al., 2023*).

The above scenarios require temperature gradients or thermal cycling. This creates degradation stress for nucleic acids and limits the scenarios to geological settings with a thermal gradient. Here, we investigated a simple and ubiquitous scenario in which a water flux through a rock pore was dried by a gas flux at constant temperature (*Figure 1*). This can be found in the vicinity of underwater degassing events, where gases percolate through rocks to reach the surface, or in porous rocks at the surface exposed to atmospheric winds (*Zhang, 2014*; *Kurnio et al., 2016*). Such a setting would be very common on volcanic islands on early Earth, which also offered the necessary dry conditions for RNA synthesis (*Powner et al., 2009*).

We created an experimental model of such an evaporation pore, shown in *Figure 1*, and studied how combined gas and water fluxes can lead to early replication of nucleic acids. We first analyzed accumulation flow speeds at the interface in *Figure 2*, then monitored cyclic strand separation dynamics in *Figure 3*, and finally showed how both drive DNA-based replication under isothermal conditions in *Figure 4*.

## Results and discussion
### Molecule accumulation at the gas-water interface

We started off by constructing a laboratory model of the rock pore shown in *Figure 1*. Here, we focused on the key properties of the system: An upward water flux evaporating at the intersection with the perpendicular gas flux. This leads to an accumulation of dissolved molecules at the interface since they cannot evaporate. Simultaneously, the momentum transfer of the gas flux induces circular currents in water, forcing molecules back into the bulk.

In the following, we analyze how these two effects act on dissolved nucleic acids. For simplicity, we used ambient air as the gas source, enabling us to focus solely on evaporation and the resulting

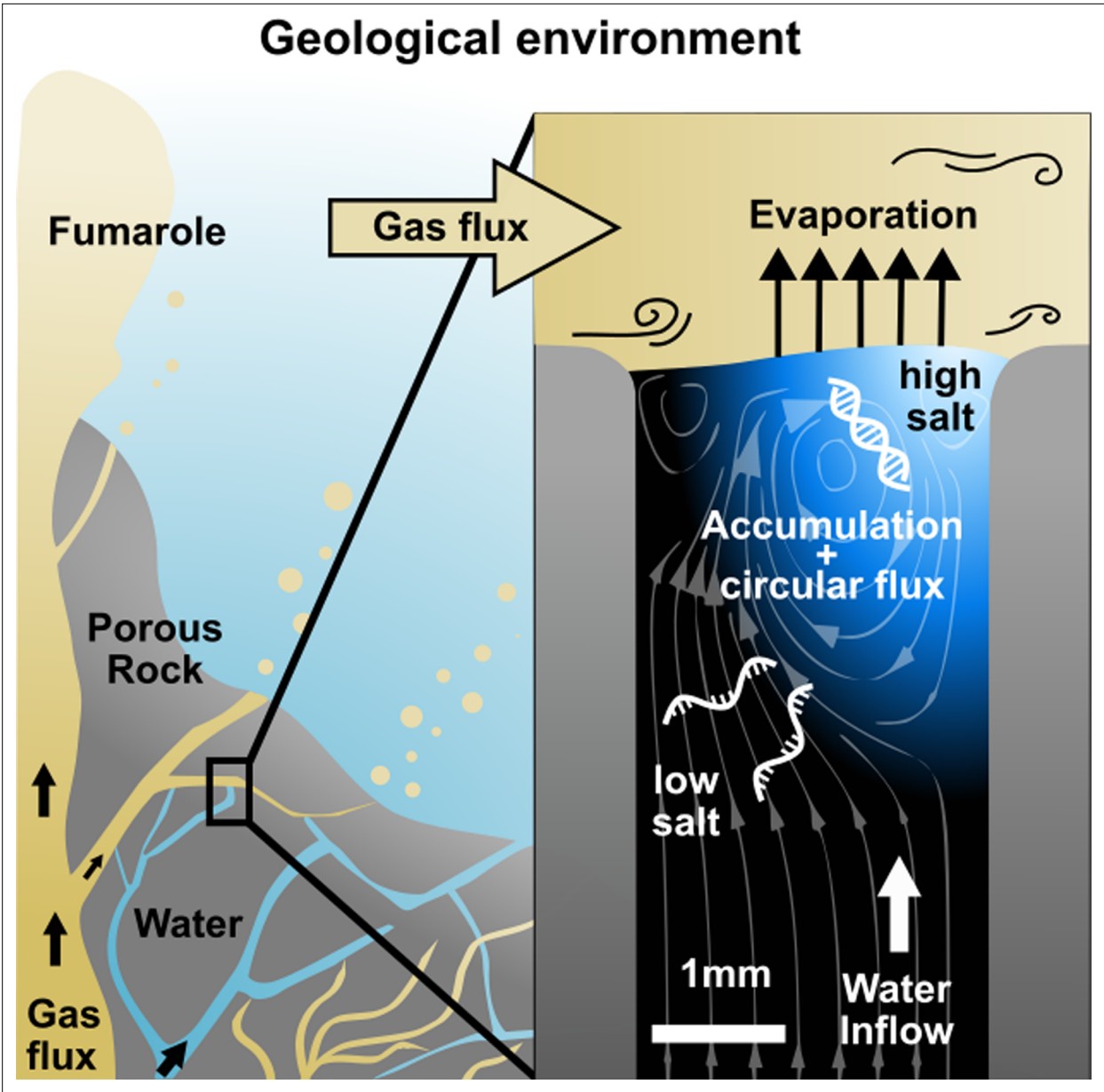

**Figure 1.** Replication at the gas-water interface. We considered a geological scenario in which water, containing biomolecules, is evaporated by a gas flow at the scale of millimeters. In volcanic porous rock, many of such settings can be imagined. The gas flow induces convective water currents and causes it to evaporate. Dissolved nucleic acids and salts accumulate at the gas-water interface due to the interfacial currents, even if the influx from below is pure water. Through the induced vortex, nucleic acids pass through different concentrations of salt, promoting strand separation and allowing them to replicate exponentially. Our experiments replicate this environment on the microscale, subjecting a defined sample volume to a continuous influx of pure water with an air flux brushing across.

currents. The velocity of the water flowing in was controlled by a syringe pump and chosen to match the velocity of the water evaporating in the given geometry. This ensured reliable and stable conditions in long-lasting experiments. For a real early Earth environment, we envision a system that self-regulates the water column's inflow by automatically balancing evaporation with capillary flows. The interface adjusts its position relative to the gas flux, moving closer if the inflow is less than the evaporation rate, or receding if it exceeds it. When the interface nears the gas flux, evaporation accelerates, while moving it away slows evaporation. This dynamic process stabilizes the system, with surface tension ultimately fixing the interface's position.

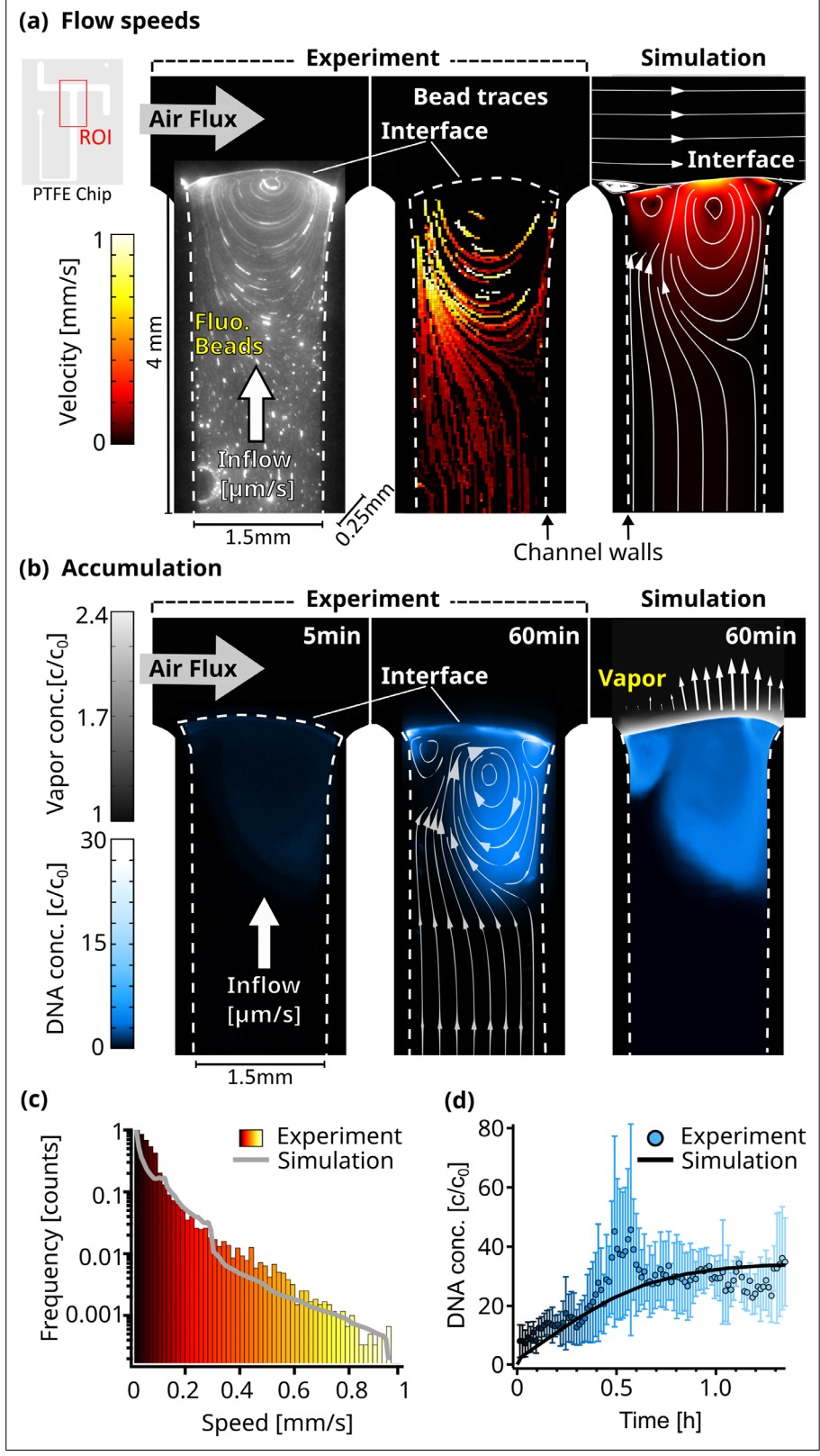

**Figure 2.** Flow and accumulation dynamics. (**a**) Imaging of fluorescent beads (0.5 µm) reveals a flow vortex right below the air-water interface, induced by the air flux across the interface (left panel). The bead movements were traced (middle panel), and the measured velocities were confirmed by a detailed finite element simulation (right panel). The PTFE chip cutout in the top left corner shows the ROI used for the micrographs. The color scale is

*Figure 2 continued on next page*

*Figure 2 continued*

equal for both simulation and experiment, and channel dimensions are 4x1.5 x 0.25 mm as indicated. Dotted lines visualize the location of the channel walls. (**b**) The accumulation of fluorescently labeled 63mer DNA was imaged and confirmed our understanding of the environment based on a diffusion model. Concentration reaches up to 30 times relative to the start c0. The accumulation profile of the experiment (middle panel) and simulation (right panel) match well, showcased by overlaying the simulated flowlines. Blue colorscale represents DNA accumulation for experiment and simulation, while grey color scale shows the relative vapor concentration in the simulation. Arrows (right panel) proportionally show the evaporation speed along the interface. (**c**) The simulated and experimentally measured distribution of flow velocities of dissolved beads plotted in a histogram, showing a similar profile. Color scale is equal to (**a**). (**d**) The maximum relative concentration of DNA increased within an hour to ≈30 X the initial concentration, with the trend following the simulation. Error bars are the standard deviation from four independent measurements.

The online version of this article includes the following video and figure supplement(s) for figure 2:

**Figure supplement 1.** Sketch of the microfluidic chamber assembly.

Between a steel frame front and an aluminium back attached to the waterbath attachment, a sapphire doublet sandwiches the teflon cutout. The 250-µm-thick cutout is connected to airflow and water flow through holes in the back sapphire. More detailed information can be found in Appendix 3.

**Figure supplement 2.** Screenshot of the user interface of the self-written LabVIEW script used for particle tracking.

**Figure supplement 3.** The geometry as it is used for the simulation and simulated velocities.

**Figure 2—video 1.** Fluorescence beads are used to track the fluid flow shown in *Figure 2a*.

https://elifesciences.org/articles/100152/figures#fig2video1

**Figure 2—video 2.** Concentration of fluorescently labeled 63mer DNA is imaged to infer the accumulation at the interface in *Figure 2b*.

https://elifesciences.org/articles/100152/figures#fig2video2

**Figure 2—video 3.** Individual repeat #1 of *Figure 2—video 2* showing raw data, underlining interfacial fluctuations.

https://elifesciences.org/articles/100152/figures#fig2video3

**Figure 2—video 4.** Individual repeat #2 of *Figure 2—video 2* showing raw data, underlining interfacial fluctuations.

https://elifesciences.org/articles/100152/figures#fig2video4

**Figure 2—video 5.** Individual repeat #3 of *Figure 2—video 2* showing raw data, underlining interfacial fluctuations.

https://elifesciences.org/articles/100152/figures#fig2video5

---

Fixed volumes of sample solutions (containing beads, labeled molecules, salts etc.) were always loaded ahead of an influx of pure water, simulating a continuous dilution scenario.

The micro scale gas-water evaporation interface consisted of a 1.5 mm wide and 250-µm-thick channel that carried an upward pure water flow of 4 nl/s ≈10 µm/s. Over this channel, a perpendicular air flow of about 250 ml/min ≈10 m/s (Appendix 3) is guided across. The temperature of the chamber was controlled by a water bath at 45 C, while a self-built fluorescence microscope provided imaging (*Figure 3—figure supplement 1*). Two-dimensional finite element simulations were performed to model the diffusion of molecules in water, as well as the flow of water and gas.

First, using a particle tracking algorithm (Appendix 4), we measured the flow velocities of individual fluorescent 0.5 µm beads to monitor the dynamics of the water flow as the air flux streamed across the interface (*Figure 2a*). As expected, these velocities were dependent on their distance from the channel walls (*Figure 2—figure supplement 3e*). The beads that were far from the interface were moving at water inflow velocities of 15 µm/s. Closer to the interface, the velocities increased to about 1 mm/s due to the momentum transfer of the gas flow (*Figure 2—video 1*). This resulted in a circular flow pattern with the vortex center right below the interface. The flow lines in *Figure 2a* show how the upward water flux reaches the interface on one side of the vortex, whereas on the opposite side, the beads are pushed back down into the bulk.

The extracted traces in *Figure 2a* were compared with a finite element simulation. In a two-dimensional projection of the experimental geometry, we modeled laminar gas and water flow, diffusive nucleic acid mass transport in water, interfacial evaporation dynamics, and momentum transfer of

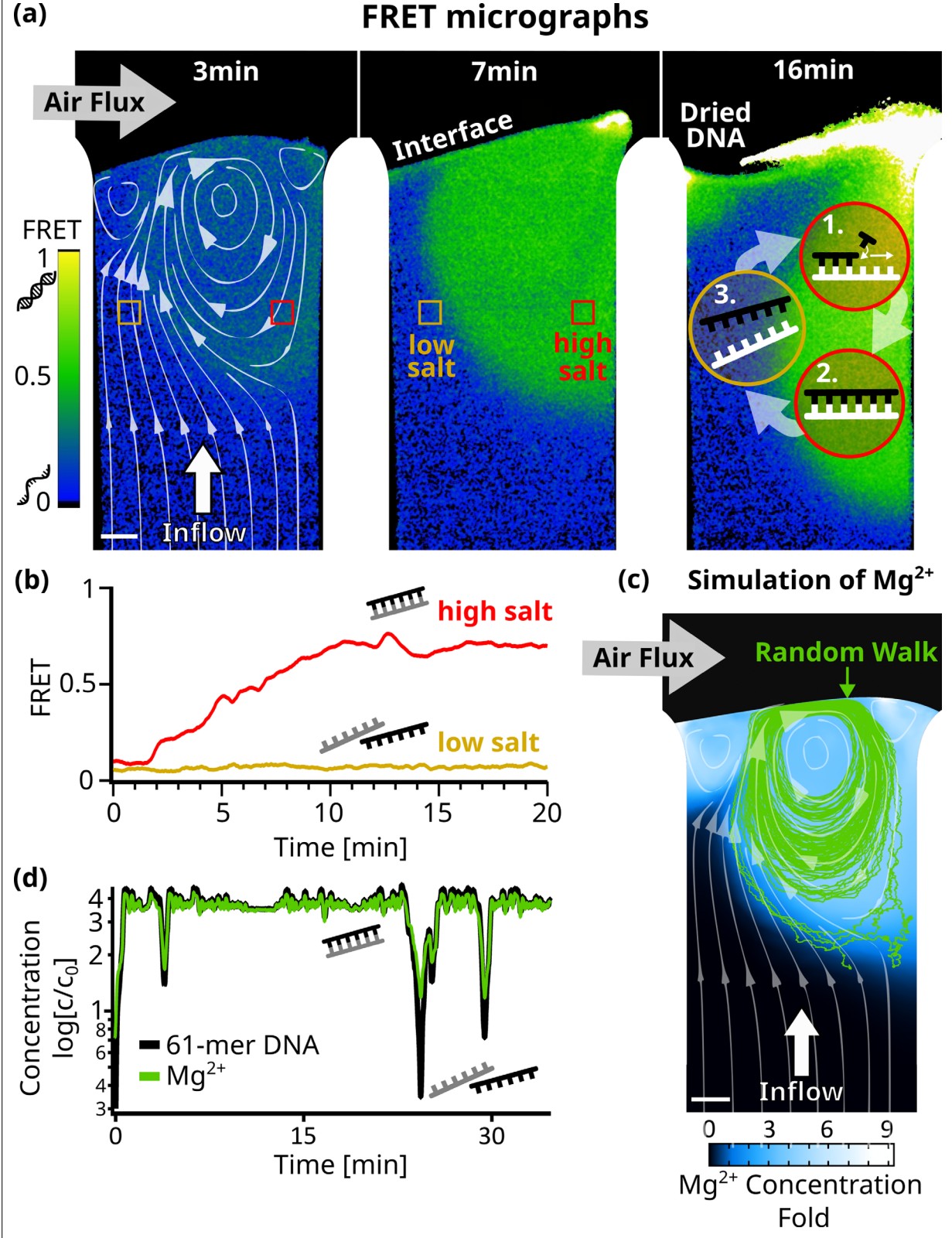

**Figure 3.** Strand separation by salt cycling. Fluorescence resonance energy transfer measurements revealed cycles of strand separation. (**a**) Micrographs of 24 bp DNA FRET pair in the chamber at 45 C. 1 μl sample (5 μM DNA, 10 mM TRIS pH7, 50 μM MgCl₂, 3.9 mM NaCl) was subjected to a 3 nl/s diluting upflow of pure water and a gas flow of 230 ml/min across. The induced vortex, shown by the simulated flow lines (left panel), overlays with regions of high FRET indicative of double-stranded DNA. The vortex flow was expected to enable replication reactions by (1+2) strand replication in the high-

*Figure 3 continued*

salt region and (3) strand separation of template and replicate in the low-salt region. Fluctuations in interface position can dry and redissolve DNA, repeatedly (see 'Dried DNA' in right panel). (**b**) FRET signals confirmed strand separation in low-salt regions and strand annealing in high-salt regions in (**a**). After about 10 min, DNA and salt accumulated at the interface ,forming stable and clearly separated regions of low – where the influx from below reaches the interface – and high – located at the vortex – FRET signals. (**c**) Comsol simulation of $Mg^{2+}$ ions $D=705\ \mu m^2/s$ in the chamber agreed with the FRET signal and showed up to ninefold salt accumulation at the interface. The path of a 61mer DNA molecule from a random walk model is shown by the green lines, and the white flowlines are taken from the simulation. (**d**) Concentrations along the DNA molecule path in (**c**) show oscillations relative to the initial concentration of up to threefold for $Mg^{2+}$ and fourfold for 61mer DNA. This could enable replication cycles, as the vortex provides high salt concentrations for replication, while drops in salt and template concentrations regularly trigger strand separation.

The online version of this article includes the following video and figure supplement(s) for figure 3:

**Figure supplement 1.** Schemes of microscopy setups used in this study.

**Figure supplement 2.** Crosstalk between donor and acceptor channel, (**a**): dd(T) and (**b**): aa(T), plotted as a function of temperature and fitted linearly.

**Figure supplement 3.** Melting curves performed in the FRET setup for NaCl and $MgCl_2$ using the 24mer strands labeled with ROX and FAM, respectively.

**Figure 3—video 1.** FRET imaging of dual-labeled DNA strands discriminates between single-stranded DNA in blue and double-stranded DNA in green to yellow, as detailed in *Figure 3a*.

https://elifesciences.org/articles/100152/figures#fig3video1

gas flowing over the water surface (Appendix 5). In agreement with the experimental results, the simulation showed a chamber-averaged water evaporation speed of 10.5 µm/s. The tangential velocity at the interface reached 0.9 mm/s. The modeled flow speed distributions agreed well with the distribution of the experimental bead velocities, as shown in *Figure 2a and c*.

To further test our understanding of the dynamics of the system, we imaged fluorescently labeled nucleic acids. The expectation was that the continuous evaporation would lead to an accumulation of the strands at the interface, while the gas flow would induce a vortex analogous to the beads. Both are found to be present in the experiment and agree qualitatively with the finite element model. In our experiment, 2 µl of 5 µM FAM-labeled 63mer DNA were introduced into the system, followed by a continuous diluting pure water inflow. Temperature, water flow, and air flow were unchanged from the previous experiment.

Water continuously evaporated at the interface, but nucleic acids remained in the aqueous phase accumulating near the interface. They could only escape downward either by diffusion or by the vortex induced by the gas flowing across the interface, pushing the molecules back deeper into the bulk (See the flow lines in *Figure 2b* taken from the simulation). As the gas flow continuously removed excess vapor, the evaporation rate remained constant. Thus, except for fluctuations, a stable interface shape should be expected. However, due to the high surface tension of water, the interface is very flexible. As the inflow and evaporation work to balance each other, the shape of the interface adjusts, likely in response to small fluctuations in gas pressure and spatial variations in water surface tension. This is leading to alterations in the circular flow fields below (*Figure 2—video 2*).

As these fluctuations are difficult to simulate, we decided to stick with one interface shape, matching evaporation and inflow speeds. The evaporation rate at the interface was therefore set to be proportional to the vapor concentration gradient and varied spatially along the interface between 5 and 10.5 µm/s (*Figure 2—figure supplement 3d*). Using the known diffusion coefficient of 95 µm²/s for the 63 mer *Mast et al., 2013*, the simulation closely matched the experimental results. In both cases, DNA accumulated in regions with circular flow patterns driven by the gas flux (*Figure 2b*, right panel).

5 min after starting the experiment, the maximum DNA accumulation was threefold, while after one hour of evaporation, around 30-fold accumulation was observed. Due to molecules residing in very shallow volumes when directly at the interface, the fluorescence signal can vary drastically compared to measurements deeper in the bulk. This can be seen in the fluctuations between independent measurements (*Figure 2—video 3*, *Figure 2—video 4*, *Figure 2—video 5*), especially around 0.5 hr shown in *Figure 2d*. The simulated maximum accumulation followed the experimental results and started saturating after about 1 hr (*Figure 2d*).

## Strand separation dynamics

As discussed earlier, strand separation is essential for the replication of nucleic acids. Only then can replication become exponential and compete with naturally exponential degradation kinetics. Usually,

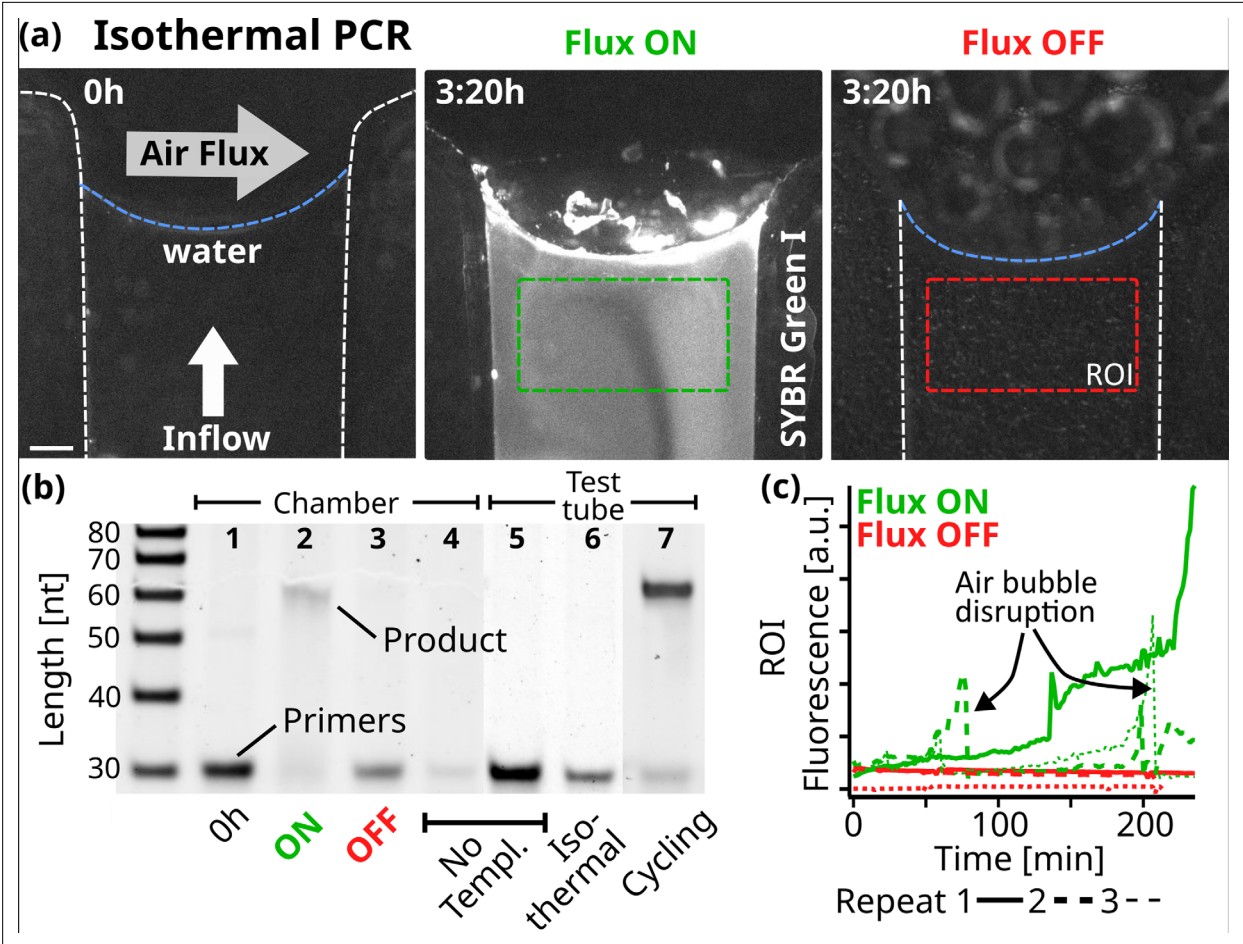

**Figure 4.** Replication. (**a**) Fluorescence micrographs of the PCR reaction in the chamber. At isothermal 68 °C, 10 µl of reaction sample was subjected to a constant 5 nl/s pure water flow toward the interface where a 250 ml/min gas flowed perpendicularly. The initial state on the left shows the background fluorescence. Fluorescence increased under flux (middle, after 3:20 hr), while without flux, the fluorescence signal remained minimal (right). The reaction sample consisted of 0.25 µM primers, 5 nM template, 200 µM dNTPs, 0.5 X PCR buffer, 2.5 U Taq polymerase, 2 X SYBR Green I. Scale bar is 250 µm. (**b**) 15% Polyacrylamide Gel Electrophoresis of the reactions and neg. controls. After 4 hr in the reaction chamber with air and water flux ON, the 61mer product was formed under primer consumption (2), unlike in the equivalent experiment with the fluxes turned OFF (3). At the beginning of the experiment (1) or in the absence of template (4), no replicated DNA was detected. The reaction mixture was tested by thermal cycling in a test tube (5-7). As expected, replicated DNA was detected only with the addition of template: (7) shows the sample after 11 replication cycles. The sample was also incubated for 4 hr at the chamber temperature (68 °C), yielding no product (6). Primer band intensity variations are caused by material loss during extraction from the microfluidic chamber. (**c**) SYBR Green I fluorescence increased when gas and water flow were turned on, but remained at background levels without flow. Fluorescence was averaged over time from the green and red regions of interest shown in (**a**). Dotted lines show the data from independent repeats. Air bubbles formed through degassing can momentarily disrupt the reaction. SYBR Green I fluorescence indicates replication, as formed products are able to hybridize.

The online version of this article includes the following video, source data, and figure supplement(s) for figure 4:

**Source data 1.** Original files for PAGE analysis displayed in *Figure 4b*.

**Source data 2.** PAGE images indicating the relevant bands displayed in *Figure 4b*.

**Figure supplement 1.** Complete schematic of the replication reaction using Taq polymerase.

**Figure supplement 2.** PAGE analysis of Taq PCR reactions.

**Figure supplement 2—source data 1.** Original files for PAGE analysis displayed in *Figure 4—figure supplement 2c and d*.

**Figure supplement 2—source data 2.** PAGE images indicating the relevant bands displayed in *Figure 4—figure supplement 2c and d*.

**Figure supplement 3.** Comparison of test-tube temperature cycling vs. chamber experiments.

**Figure supplement 3—source data 1.** Original files for PAGE analysis displayed in *Figure 4—figure supplement 3a*.

**Figure supplement 3—source data 2.** PAGE images indicating the relevant bands displayed in *Figure 4—figure supplement 3a*.

*Figure 4 continued on next page*

*Figure 4 continued*

**Figure 4—video 1.** Top fraction of the chamber used for the figures.
https://elifesciences.org/articles/100152/figures#fig4video1

**Figure 4—video 2.** Individual repeat #1 of *Figure 4—video 1*.
https://elifesciences.org/articles/100152/figures#fig4video2

**Figure 4—video 3.** Individual repeat #2 of *Figure 4—video 1*.
https://elifesciences.org/articles/100152/figures#fig4video3

**Figure 4—video 4.** Whole length of the chamber.
https://elifesciences.org/articles/100152/figures#fig4video4

---

an elevation of temperature can separate strands but is accompanied by a higher risk for hydrolysis. The chosen isothermal setting requires changes in salt concentration for this process. More specifically, the circular fluid flow at the interface provided by the gas flux, together with Brownian motion, was expected to drive cyclic strand separation by forcing nucleic acid strands through areas of varying salt concentrations.

We used Förster resonance energy transfer (FRET) microscopy to optically measure the strand separation of DNA (*Figure 3*). A high FRET signal indicates that two DNA strands are bound, while a low FRET signal indicates that the strands are separated. In this way, FRET becomes an indirect measure of the salt concentration, since a low-salt concentration will induce strand separation due to the reduced ionic shielding of the charged DNA or RNA backbones. Specifically, we chose a complementary 24mer DNA pair, with the FRET-pair fluorophores positioned centrally on opposite strands. 1 µL Sample (10 mM TRIS at pH 7, 50 µM MgCl$^2$, 3.9 mM NaCl, and 5 µM of each DNA strand) was injected into the chamber and flushed toward the interface by pure water with all other conditions equal to before.

*Figure 3a* shows micrographs of the recorded FRET values for each pixel (*Figure 3—video 1*). Initially, the FRET signal increased near the interface (green), indicating areas where DNA is forming double-stranded DNA. This area is localized around the vortex created by the gas flow across the interface. In the upward flow to the left of the vortex, DNA was found to be single-stranded (blue). During the course of the experiment, the low and high FRET regions remained stably separated (*Figure 3b*). This configuration suggests that the vortex could drive a cycle of replication and strand separation (see the scheme in *Figure 3a* - right panel).

To confirm this, we simulated the accumulation of $Mg^{2+}$ ions in the chamber (Appendix 6), since divalent ions have a large effect on the melting temperature of nucleic acids (*Schildkraut and Lifson, 1965*). We then used a Monte Carlo random walk model (Appendix 7) to simulate individual 61mer DNA molecules following the vortex and undergoing Brownian motion. Such a path is shown in *Figure 3c*, plotted over the simulated steady-state concentration of $Mg^{2+}$ along with the simulated flow lines. Starting in a region of low $Mg^{2+}$ concentration, the strand enters the vortex created by the gas flow. We have plotted the $Mg^{2+}$ concentration along its path, showing significant salt oscillations of up to 3 X the initial salt concentration, capable of inducing strand separation (*Figure 3d*, *Figure 3—figure supplement 3a*). Rayleigh-Bénard convection cells generate similar patterns to those seen in *Figure 3c*. The oscillations in salt concentration resemble the temperature fluctuations observed in convection-based PCR reactions from earlier studies (*Muddu et al., 2011*; *Braun et al., 2003*), which also showed that chaotic temperature variations like the salt variations in our system, even enhanced the efficiency of the PCR reaction, compared to periodic ones.

In the experimental conditions used here, RNA would also not readily degrade, even if the strand enters the high salt regimes (Appendix 8). Using literature values for hydrolysis rates under the deployed conditions, we estimate dissolved RNA to have a half-life of around 83 days.

When plotting the simulated steady-state concentration of other dissolved – complementary – 61mer DNA molecules along its path, we observed even stronger oscillations of up to 4 X the initial concentration. Together with significant drops in $Mg^{2+}$ concentration, this suggests the possibility of exponential replication by strand separation cycles.

## Isothermal replication with PCR

We saw that nucleic acids and salts accumulated near the interface, but far from the interface, in the bulk below, the concentrations remained vanishingly low due to the diluting inflow of pure water. The

air flux induced an accumulation pattern of vortices in which molecules were trapped. The salt and DNA concentration changed cyclically, resulting in periodic strand separation of nucleic acids. Motivated by the above results, we used a model system to test whether nucleic acid replication could actually be implemented in this environment.

We chose to use Taq DNA Polymerase because it does not have a protein-based strand-separating mechanism. Starting with a 51mer template and two 30mer primer strands, each with a 5'-AAAAA overhang for detection, the reaction is expected to form a 61mer replicate (Appendix 9), the same length as the DNA used in the random walk model in *Figure 3c and d*. In contrast to standard PCR, which uses thermal cycling to separate the strands, we operated the experiment at isothermal conditions (68 °C) and used 10 µl of the reaction mix (0.25 µM primers, 5 nM template, 200 µM dNTPs, 0.5 X PCR buffer, 2.5 U Taq polymerase, 2 X SYBR Green I). This reaction mixture was then exposed to a constant pure water influx of 5 nl/s toward the gas-water interface, matching the rate of evaporation at the interface.

Through the oscillations in salts and DNA observed along the random walk, we expected the 61mer product strand to be able to separate from its respective template strand, enabling exponential replication. The progress was monitored using the intercalating dye SYBR Green I, which binds preferentially to double-stranded (*Dragan et al., 2012*). *Figure 4a* shows fluorescence micrographs of the reaction in the chamber. Initially, minimal fluorescence is seen, indicating that the replicated templates are below the detection limit of SYBR Green. *Figure 4c* shows how the SYBR Green fluorescence increased after two hours in the displayed region of interest (ROI), recording the increase of replicated DNA forming duplex structures. In other repetitions of the reaction, this increase was sometimes even observed earlier, around the 1-hr mark (dotted lines). However, air bubbles nucleated by degassing events rise and temporarily dry out the channel, interrupting the reaction until the liquid refills the channel (*Figure 4—video 1*, *Figure 4—video 2*, *Figure 4—video 3*, *Figure 4—video 4*). Despite our best efforts, we were unable to fully prevent this, especially given the high temperatures required for Taq polymerase activity. In an identical setting when the gas and water flux were switched off, no fluorescence increase was found (See *Figure 4c* red lines). Fluorescence variations are additionally caused by fluctuations in the position of the gas-water interface, as discussed earlier.

Replication was confirmed under flux with the 61mer product being visible in gel electrophoresis with depleted primers (*Figure 4b*). With both gas flow and water influx turned off, no product band was found. We verified the replication reaction by repeating the experiments without the addition of the template, primer, or DNA in the chamber as well as in a test tube. *Figure 4—figure supplement 2* shows all independent repeats of the corresponding experiments. No product was detected in any of these cases, ruling out reaction limitations such as primer dimer formation. Primer dimers would form even in the absence of a template strand and would be identifiable through gel electrophoresis. As Taq polymerase requires a significant overlap between the two dimers to bind, this would result in a shorter product compared to the 61mer used here. We also compared the chamber experiment with a regular, temperature cycling-based, PCR reaction in a test tube, revealing that in the chamber, about 10–11 cycles of PCR were finished after the 4 hr of experiment (Appendix 9). The findings above confirm that the gas flow at the simulated rock opening was necessary for nucleic acid replication.

## Conclusion

In this work, we investigated a prebiotically plausible and abundant geological environment to support the replication of nucleic acids. We considered an isothermal setting of gas flowing over an open rock pore filled with water. Previously, thermal gradients have been used to separate the strands of nucleic acids, risking their degradation. Now, the combined gas and water flow at an open pore triggers salt oscillations. We found that this condition supports oligonucleotide replication. We began by probing the system with fluorescent bead and DNA measurements, finding our results to agree with fluid dynamics theory using finite element simulations.

While DNA accumulates at the vortex close to the interface, oscillations in nucleic acid and salt concentration are created by a combination of molecular accumulation and interfacial flow, periodically separating nucleic acid strands under chemically gentle conditions. Due to the limitations of RNA-based replication, we probed the environment with protein-driven DNA replication and found isothermal replication in this common geological micro-environment, showing that it provides a setting for early nucleic acid replication chemistry.

Prebiotic chemical reactions such as polymerization of imidazole (*Szostak, 2012*) or 2′,3′-cyclic phosphate-activated nucleotides (*Dass et al., 2023*; *Serrão, 2023*) will benefit from the reduced RNA hydrolysis in the gentle isothermal replication environment. Most importantly, the combination of actively generated high concentrations by evaporation, dry-wet cycles at the interface caused by interface fluctuations, shielding from UV damage, and the possibility of constant feeding by water influx makes the environment a compelling candidate for implementing the geophysical boundary condition of the early RNA world stage of emergent life.

Furthermore, we expect that other gases, such as $CO_2$, could establish chemical gradients in this environment. Such gradients have been observed in thermal gradients before *Keil et al., 2017*, and finding similar behavior in an isothermal environment would be a significant discovery. Physical non-equilibria, such as steep temperature gradients, pose many boundary conditions, decreasing the likelihood of readily finding such a setting on early Earth. This isothermal environment, however, greatly extends the repertoire of prebiotic settings that enable replication on early planets.

## Materials and methods

A microfluidic chamber was created between two sapphire plates (0.5 mm thick on the front and 1 mm thick on the back), sandwiching a 0.25-mm-thin Teflon sheet that defined the geometry created by a computer-controlled cutter. The plates were held together by a steel frame bolted to an aluminum back to ensure gas tightness. The back was connected to a water bath (Julabo) to control the temperature. Samples were injected into the chamber using syringe pumps (Nemesys) with tubings inserted into holes in the back sapphire. Gas flow was generated under pressure control using the AF1 dual pump system (Elveflow). Temperature was measured during the experiments with a thermal sensor attached to the back sapphire. A more detailed schematic of the microfluidic chamber can be found in Appendix 3.

DNA fluorescence measurements were performed in a self-built tilted epi-fluorescence microscope setup using two M490L4 and M625L3 light-emitting diodes (Thorlabs), a 470/622 H dual-band excitation filter (AHF), a 497/655 H dual-band dichroic mirror, and a 537/694 H dual-band emission filter. A more detailed schematic of the setup can be found in Appendix 2. DNA strands were ordered from biomers.net, including purification by high-performance liquid chromatography (Appendix 1). The strands used for fluorescence quantification of accumulation are (5′–3′)–24 bp DNA: *CY5*CGTA GTAAATATCTAGCTAAAGTG, 63 bp DNA: *FAM*CCAGCCTCCAGTGCCTCGTATCATTGTGCCAA AAGGCACAATGATACGAGGCACTGGAGGCTG (*Appendix 1—table 1*) diluted to 5 µM in Nuclease-free water. Images were captured using a Stingray F145B camera (Allied Vision). Bead experiments used 0.5 µM fluorescent microspheres (Invitrogen) diluted 1/2000 in water (Appendix 4).

2D finite element simulations were performed using COMSOL Multiphysics 5.4. Fluid dynamics were simulated by solving the Navier-Stokes equation in two dimensions. Parameters used are available in *Appendix 6—table 1*. The complete description of the model can be found in Appendix 5.

FRET imaging was performed using a second custom-built fluorescence microscopy setup consisting of light-emitting diodes (M470L2, M590L2; Thorlabs) combined by a dichroic mirror on the excitation side, while an Optoplit II with a ratiometric filter set (DC 600LP, BP536/40, BP 630/50) and a Stingray-F145B ASG camera (Allied Vision Technologies) through a 1 X objective (AC254 100 A-ML Achromatic Doublet; Thorlabs) detected and superimposed both fluorescence emission channels (Appendix 2). The DNA sequences used for FRET experiments were: strand 1 5′-CGTAGTAAATAT*FAM*CTAGCTAA AGTG-3′, strand 2 5′-CACTTTAGCTAGAT*ROX*ATTTACTACG-3′ (*Appendix 1—table 1*). The two labeled complementary strands were diluted from stock solution (100 µM in nuclease-free water) and mixed together to a final concentration of 5 µM in buffer (10 mM TRIS, 50 µM $MgCl_2$, 3.9 mM NaCl, pH7). To promote annealing of the two complementary strands, the solution was heated and slowly cooled from 80°C to 4°C (ramp rate of –1 C per 5 s) in a standard thermocycler (Bio-Rad CFX96 Real-Time System) prior to each experiment.

PCR was performed using an AllTaq PCR Core Kit (QIAGEN). Samples were mixed with 0.5 X AllTaq PCR Buffer, 5 nM template strand, 0.25 µM primers, 200 µM of each dNTP, 2 X SYBR Green I and AllTaq polymerase at 2.5 U/reaction. The reaction in the thermocycler was performed using a temperature protocol of 95 °C for 2 min for heat activation of the enzyme, then annealing the primers to 52 °C for 10 s, then 68 °C for 10 s, and finally 10 s at 95 °C. This cycle was repeated 40 times (*Figure 4—figure supplement 2b*). The reaction in the chamber was performed with 10 µl

of the above mixture at 68 °C. The solution was also heat activated at 95 °C for 2 min followed by an annealing step to 52 °C before loading into the chamber. The DNA sequences for the reaction were as follows: Template (5'–3')–51 bp DNA: TTAGCAGAGCGAGGTATGTAG-GCGGGACGCTCA GTGGAACGAAAACTCACG, Reverse primer (5'–3')–30 bp DNA: AAAAACGTGAGTTTTCGTTCCACT GAGCGT, forward primer (5'–3')–30 bp DNA: AAAAATTAGCAGAGCGAGGTATGTAGGCGG (see also *Appendix 1—table 1*).

For PAGE and gel imaging, a 15% denaturing (50% urea) polyacrylamide gel with an acrylamide:bis ratio of 29:1 was solidified with TEMED (tetramethylethylenediamine) and ammonium persulfate. 2 µl of sample was mixed with 7 µl of 2 X loading buffer (Orange G, formamide, EDTA), of which 5 µl were loaded onto the gel. Staining was performed with 2 X SYBR Gold in 1 X TBE buffer for 5 min and the gel was imaged using the ChemiDOC MP imaging station (Bio-Rad).

## Acknowledgements

We would like to acknowledge the following agencies for funding: Deutsche Forschungsgemeinschaft (DFG, German Research Foundation) – Project-ID 364653263 – CRC 235, Deutsche Forschungs-gemeinschaft (DFG, German Research Foundation) – Project-ID 521256690 – CRC 392, Deutsche Forschungsgemeinschaft (DFG, German Research Foundation) – Project-ID 201269156 – SFB 1032, Volkswagen Initiative' Life? – A Fresh Scientific Approach to the Basic Principles of Life', HFSP RGP003/2023, Germany's Excellence Strategy EXC-2094–390783311, Simons Foundation #327125, European Research Council EvoTrap #787356, ERC-2017-ADG. This work was supported by the Center for Nanoscience Munich (CeNS).

## Additional information

### Funding

| Funder | Grant reference number | Author |
|---|---|---|
| Deutsche Forschungsgemeinschaft | CRC 235 | Philipp Schwintek<br>Emre Eren<br>Christof Bernhard Mast<br>Dieter Braun |
| Deutsche Forschungsgemeinschaft | CRC 392 | Philipp Schwintek<br>Emre Eren<br>Christof Bernhard Mast<br>Dieter Braun |
| Deutsche Forschungsgemeinschaft | SFB 1032 | Philipp Schwintek<br>Emre Eren<br>Christof Bernhard Mast<br>Dieter Braun |
| Volkswagen Foundation | Life? - A Fresh Scientific Approach to the Basic Principles of Life' | Philipp Schwintek<br>Emre Eren<br>Christof Bernhard Mast<br>Dieter Braun |
| Human Frontier Science Program | HFSP RGP003/2023 | Philipp Schwintek<br>Emre Eren<br>Christof Bernhard Mast<br>Dieter Braun |
| Simons Foundation | Simons Collaboration on the Origins of Life #327125 | Philipp Schwintek<br>Emre Eren<br>Christof Bernhard Mast<br>Dieter Braun |
| European Research Council | Advanced Grant EvoTrap #787356 ERC-2017-ADG | Philipp Schwintek<br>Emre Eren<br>Christof Bernhard Mast<br>Dieter Braun |

| Funder | Grant reference number | Author |
|--------|------------------------|--------|
| Center for NanoScience, Ludwig-Maximilians-Universität München | | Philipp Schwintek<br>Emre Eren<br>Christof Bernhard Mast<br>Dieter Braun |
| Deutsche Forschungsgemeinschaft | Germany's Excellence Strategy EXC-2094-390783311 | Philipp Schwintek<br>Emre Eren<br>Christof Bernhard Mast<br>Dieter Braun |
| Deutsche Forschungsgemeinschaft | Project-ID 364653263 | Philipp Schwintek<br>Emre Eren<br>Christof Bernhard Mast<br>Dieter Braun |
| Deutsche Forschungsgemeinschaft | Project-ID 521256690 | Philipp Schwintek<br>Emre Eren<br>Christof Bernhard Mast<br>Dieter Braun |
| Deutsche Forschungsgemeinschaft | Project-ID 201269156 | Philipp Schwintek<br>Emre Eren<br>Christof Bernhard Mast<br>Dieter Braun |
| Deutsche Forschungsgemeinschaft | Origins Cluster | Philipp Schwintek<br>Emre Eren<br>Christof Bernhard Mast<br>Dieter Braun |

The funders had no role in study design, data collection and interpretation, or the decision to submit the work for publication.

### Author contributions

Philipp Schwintek, Conceptualization, Resources, Data curation, Software, Formal analysis, Validation, Investigation, Methodology, Writing – original draft, Project administration; Emre Eren, Formal analysis, Investigation, Methodology; Christof Bernhard Mast, Conceptualization, Resources, Software, Validation, Investigation, Project administration, Writing – review and editing; Dieter Braun, Conceptualization, Resources, Data curation, Formal analysis, Supervision, Funding acquisition, Validation, Investigation, Visualization, Methodology, Project administration, Writing – review and editing

### Author ORCIDs
Philipp Schwintek ⬤ https://orcid.org/0000-0002-6440-5918
Dieter Braun ⬤ https://orcid.org/0000-0001-7751-1448

Reviewer #1 (Public review): https://doi.org/10.7554/eLife.100152.3.sa1
Reviewer #2 (Public review): https://doi.org/10.7554/eLife.100152.3.sa2
Author response https://doi.org/10.7554/eLife.100152.3.sa3

---

# Additional files

### Supplementary files
MDAR checklist

### Data availability
Figure 4—source data 1, Figure 4—figure supplement 3—source data 1, and Figure 4—figure supplement 2—source data 1 contain original blots used for PAGE analysis. Raw images used in this study, as well as individual data points are available in the data repository: https://doi.org/10.5282/ubm/data.621. Labview code used in Figure 3 can also be found in the data repository: https://doi.org/10.5282/ubm/data.621. The COMSOL Multiphysics simulation file used throughout the manuscript is also in the data repository: https://doi.org/10.5282/ubm/data.621.

The following dataset was generated:

| Author(s) | Year | Dataset title | Dataset URL | Database and Identifier |
|---|---|---|---|---|
| Schwintek P, Eren E, Braun D | 2025 | Data repository for Schwintek et al. in "Prebiotic Gas Flow Environment Enables Isothermal Nucleic Acid Replication" | https://doi.org/10.5282/ubm/data.621 | Open Data LMU, 10.5282/ubm/data.621 |

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

## Appendix 1

## DNA strands

**Appendix 1—table 1.** DNA sequences as ordered from biomers.net.

| Length | 5'- Sequence –3' | Label |
|---|---|---|
| 63mer | CCAGCCTCCAGTGCCTCGTATCATTGTGCCAA AAGGCACAATGATACGAGGCACTGGAGGCTG | 5' FAM |
| 24mer FRET strand 1 | CGTAGTAAATA8CTAGCTAAAGTG | 8=FAM |
| 24mer FRET strand 2 | CACTTTAGCTAGA8ATTTACTACG | 8=ROX |
| 51mer Template | TTAGCAGAGCGAGGTATGTAGGCG GGACGCTCAGTGGAACGAAAACTCACG | - |
| 30mer forward primer | AAAAA TTA GCA GAG CGA GGT ATG TAG GCG G | - |
| 30mer reverse primer | AAAAA CGT GAG TTT TCG TTC CAC TGA GCG T | - |

## Appendix 2

### Experimental setups

All setups used in this paper were designed and assembled from components purchased from several companies. *Figure 3—figure supplement 1* shows schematics of both setups used. All specifications below are given in nanometers.

Setup A was built to capture fluorescence signals from beads and labeled DNA and was constructed using SM1 lens tubes (Thorlabs), excitation LEDs M490L4 and M625L3-C4 (Thorlabs), excitation filters (470/622 H, AHF) and emission filters (497/655 H, AHF), and a dual-band dichroic beamsplitter (497/655 H, AHF). The camera (Stingray F-145 B/C) was purchased from Allied Vision. An achromatic doublet (AC254-100-A-ML, Thorlabs) was used as the objective. Setup B was built to capture FRET. A Zeiss Axiotech Vario microscope body was used and equipped with excitation LEDs M590L4 (yellow) and M470L2 (blue) (Thorlabs) with excitation filters BP588/20 and BP482/29 (Thorlabs) coupled into the same beamline via a DC R 475/40 beamsplitter. A dual-band dichroic mirror (505/606T) and an Optosplit II with a ratiometric filter set (DC600 LP, BP630/50 and BP 536/40) were coupled to the system to capture the FRET signal of the dyes ROX and FAM. Camera and objective are identical to setup A.

For both setups, fluorescence measurements were performed under curtains to ensure low background noise. Temperature was controlled by a JULABO Corio CD water bath connected to the microfluidic chamber (Appendix 3) by thermally insulated tubing. The temperature was measured directly at the back sapphire surface of the chamber. Gas flow was generated using an AF1 Dual Pump (Elveflow) system with ambient air as the gas source, and gas flow rate was measured with a flow sensor (FS2000, from IDT). Syringes were driven by a Cetoni syringe pump (Nemesys).

## Appendix 3

### Microfluidic chamber

The microfluidic chamber used for the experiments in this work was constructed as follows:

Two 60x22 mm sapphire plates (front 1 mm thick, back 0.5 mm thick) were used to sandwich a 250-μm-thick Teflon foil from which the channels were cut using a Graphtec CE6000-40 Plus plotter. We used sapphire plates for their higher thermal conductivity compared to normal glass. The back sapphire had four holes to allow gas and water flow in and out of the chamber. The sandwich was held together by the steel frame and aluminum back, which were screwed together with a torque of 0.2 Nm. The aluminum back was held in place on the water bath fixture by magnets. A thin (25μm) graphite foil was placed between the sapphire back and the aluminum back to increase the thermal conductivity. *Figure 2—figure supplement 1* shows an exploded view of the chamber.

## Appendix 4

### Bead measurements

A self-written Labview script (*Figure 2—figure supplement 2*) was used to track the movement of the fluorescent beads. 10 µl solution containing 0.5 µm fluorescent beads (Invitrogen, Eugene, Oregon, USA, Lot: 31,373 W), diluted 1–2000,, were loaded into the chamber and subjected to a pure water upward flow of 3 nl/s and a perpendicular gas flow of 230 ml/min (*Figure 2—video 1*). Images were captured using a 50ms exposure time, resulting in approximately 20 fps. Our setup did not allow very fast beads to be traced, as the maximum frame rate of 20 fps did not capture particles faster than about 1 mm/s. The 2D map of traced velocities therefore has some dark spots near the interface where the fastest beads are located.

## Appendix 5

### Finite element simulations

Finite element simulations were performed using COMSOL Multiphysics 5.4. A 2D geometry was designed using the same parameters as the Teflon cutout for the experimental chamber. Since the dynamics of the experiment take place mainly in the x-y plane, the system was simulated without the z dimension, focusing on the key properties. The geometry is coupled to a gas inlet and outlet as well as a water inlet from the bottom, analogous to the experiment (Appendix 5). The normal gas inflow has been set to experimentally measured values (about 236 ml/min), resulting in velocities up to about 12 m/s. The gas outlet uses pressure as its boundary condition, releasing as much gas as necessary to maintain a constant pressure. The system is assumed to be laminar, since the velocities don't exceed Mach <0.3. The transport of water vapor in the gas is coupled at the top, using the stationary velocity field previously established. Simultaneously, the stationary velocity field of the laminar upward flow of water was calculated and coupled with the time-dependent transport of dilute species, in this case dissolved DNA or salts. For both the gas and the water, Navier-Stokes equations were solved under the assumption that the flows are laminar due to their relatively low speed to viscosity ratio (Reynolds number). The flows can therefore be considered incompressible, the density constant, and the continuity equation reduces to the condition:

$$\nabla \cdot \boldsymbol{u} = \boldsymbol{0}. \tag{1}$$

with $\rho$ denoting the mass density and $u$ the velocity field. The Navier-Stokes equation then reduces to

$$\rho(\boldsymbol{u} \cdot \nabla)\boldsymbol{u} = \nabla \cdot [-p\boldsymbol{I} + \eta(\nabla \boldsymbol{u}) + (\nabla \boldsymbol{u})^T] + \boldsymbol{F} = \boldsymbol{0}, \tag{2}$$

with $p$ being pressure, $\boldsymbol{I}$ the unity tensor, $\eta$ the fluid dynamic viscosity and $\boldsymbol{F}$ the external forces applied to the liquid. The reference pressure was set to 1[atm], the reference temperature was 45 °C and all surfaces, except the gas-water interface, are described as non-slip boundary conditions. The diffusion-dependent transport of diluted species was simulated using Fick's law and convection due to the laminar flow fields:

$$\frac{\delta c_i}{\delta t} + \nabla \cdot (-D_i \nabla c_i + \boldsymbol{u}i) = 0 \tag{3}$$

Equivalently, *Equations 2; 3* are used for the dynamics of the gas channel. The boundary condition to combine the gas flow with the water flow is embedded in the gas-water interface: The velocity field components in x- and y-direction of the gas as well as water flow are required to be equal at the gas-water interface:

$$\boldsymbol{u} = \boldsymbol{u}2 \tag{4}$$

where $\boldsymbol{u}$ describes the vector field of water, while $\boldsymbol{u}2$ denotes the vector field of the passing gas. The interface acts as a sliding wall, moving in the x-direction of the gas flux to emulate the momentum transfer of the wind to the water surface. To simulate the evaporation of water into the gas phase, we used the August equation, describing the relation between saturation vapor pressure and temperature:

$$P_{sat} = \exp 20.386 - \frac{5132K}{T}[mmHg] \tag{5}$$

The saturation concentration of water vapor therefore is

$$c_{sat} = \frac{P_{sat}}{R \cdot T} \tag{6}$$

,where R denotes the ideal gas constant and T the temperature. At the interface, the boundary condition reads:

$$c_{vapor} = c_{sat}, \tag{7}$$

while at the ceiling of the gas channel, far away from the interface, the concentration is set to:

$$c_{vapor} = h \cdot c_{sat}, \tag{8}$$

where h denotes the relative humidity in percent. Furthermore, the speed of evaporation at the interface is proportional to the vapor concentration gradient:

$$\nu_{\mathrm{evap}} = -D_{vap} \cdot \frac{M}{\rho} \cdot \nabla c_{vap} \tag{9}$$

where M represents the molar mass of water, $\rho$ the density of water and $D_{vap}$ the diffusion coefficient of vapor.

**Appendix 5—table 1.** Parameters used for the finite elements simulation.
Final set of parameters used to simulate the system. Water-specific parameters such as dynamic viscosity or density were taken from inbuilt features of COMSOL Multiphysics 5.

| Parameter | Value | Description |
|---|---|---|
| $D_{vapor}$ | (21.2E-6)*(1/1[K])*(1 + (0.0071*(T - 273))) [m²/s] | Diffusion Vapor |
| $D_{63mer}$ | 643*n$^{-0.46}$[µm²/s]=95.6 µm²/s | Diffusion Coefficient of a 63mer DNA Strand *Mast et al., 2013* |
| $D_{Mg^{2+}}$ | 705 µm²/s | Diffusion Coefficient of Mg *Yuan-Hui and Gregory, 1974* |
| $c(vapor)_0$ | humidity*0.01*(exp(20.386-(5132[K]/T))[mmHg]) / (R * T) | Initial Vapor concentration |
| Humidity | 40 % | Ambient relative humidity |
| $p_{sat}$ | (exp(20.386 - (5132[K]/T))[mmHg]) | Vapor saturation pressure |
| $c_{sat}$ | $p_{sat}$/(R*T) | Vapor saturation concentration |
| $M_{vapor}$ | 0.0180 [kg/mol] | Molar mass of vapor |
| $M_{water}$ | 18.01528 [g/mol] | Molar mass of water |
| T | 45 °C | Temperature |

## Appendix 6

### Förster resonance energy transfer (FRET)

To measure the FRET signal in our microscope, we used an alternating illumination protocol (*Figure 3—video 1*). The FRET pair FAM-ROX was excited by two LEDs in rapid succession. The blue LED excited the donor dye (FAM), while the acceptor (ROX) can only be excited indirectly while both dyes are in the FRET region. The yellow LED excited only the acceptor dye (ROX). Individual images of each illumination were captured using an Optosplit II to separate the individual emission wavelengths of FAM and ROX before they reached the camera. This allowed the emission of FAM and ROX to be captured simultaneously for each of the two illuminations, providing four images for each time point: DD, DA, AA, and AD (see *Appendix 6—table 1* for details). The spatially averaged, temperature-dependent, crosstalk- and artifact-corrected FRET signal was calculated using *Equation 10*; *Mast et al., 2013*. Crosstalk between the two channels (aa(T) and dd(T)) was calculated in separate experiments using the same setup parameters with the *Equations 11; 12*, respectively. The data used for the crosstalk calculations are shown in *Figure 3—figure supplement 2*. To test how different salt concentrations affect the FRET signal, we performed melting curves of different salt concentrations (*Figure 3—figure supplement 3*). We found that the sodium concentration has little effect on the melting temperature, while $Mg^{2+}$ strongly influences the hybridization state. In the FRET experiment in *Figure 3*, the initial $Mg^{2+}$ concentration was 50 µM at 45 °C. In this state the, double-stranded fraction is about 0.3. When the salts accumulated at the interface, salt concentrations increased up to ninefold, strongly changing the double-stranded fraction to around 0.8.

$$FRET(T) = \frac{DA(T) - dd(T) \cdot DD(T) - aa(T) \cdot AA(T)}{AA(T)} \tag{10}$$

where dd(T) and aa(T) represent the non-FRET artifacts (crosstalk) in the DA and AA channels and are defined as:

$$dd(T) = \frac{DA_D(T)}{DD_D(T)} \tag{11}$$

and

$$aa(T) = \frac{D_{AA}(T)}{A_{AA}(T)} \tag{12}$$

Before each experiment, a melting curve of the FRET strands was performed inside the experimental setup chamber. The melting curve was used to normalize the FRET signal to 0 and 1 using the following equation:

$$FRET_{norm}(T) = \frac{FRET(T) - \alpha}{\beta} \tag{13}$$

where $\alpha = min(FRET(T))$ and $\beta = max(FRET(T))$.

**Appendix 6—table 1.** Channel definitions for FRET calculation.
First capital letter denotes the excitation wavelength (D=Donor, A=Acceptor), second the measured emission wavelength and the subscript stands for the label used in a separate experiment to determine crosstalk-related artifacts.

| Channel | Excitation | Emission | Label |
|---------|------------|----------|-------|
| DD | FAM - 470 nm | FAM - 536 nm | FAM/ROX |
| DA | FAM - 470 nm | ROX - 630 nm | FAM/ROX |
| AA | ROX - 590 nm | ROX - 630 nm | FAM/ROX |
| AD | ROX - 590 nm | FAM - 536 nm | FAM/ROX |
| $AA_A$ | ROX - 590 nm | ROX - 630 nm | ROX |

*Appendix 6—table 1 Continued on next page*

*Appendix 6—table 1 Continued*

| Channel | Excitation | Emission | Label |
|---|---|---|---|
| $DA_A$ | FAM - 470 nm | ROX - 630 nm | ROX |
| $DD_D$ | FAM - 470 nm | FAM - 536 nm | FAM |
| $DA_D$ | FAM - 470 nm | ROX - 630 nm | FAM |

## Appendix 7

### Random walk model

For the random walk model, we first used the existing Comsol simulation and ran the simulation with $Mg^{2+}$ ions and a 61mer DNA as the dilute species (the diffusion constant of 705 $\frac{\mu m^2}{s}$ for $Mg^{2+}$ at 25 °C was taken from *Yuan-Hui and Gregory, 1974*, and the diffusion constant of 97.04 $\frac{\mu m^2}{s}$ for a 61mer DNA strand from *Mast et al., 2013*). The simulation was performed in the same chamber, with the same characteristics and settings as in Appendix 5. The resulting stationary salt and DNA concentration fields after 2 hr were exported as a 2D table with 200 values in x-direction and 200 values in y-direction, representing the whole simulated geometry. The same was done for the stationary laminar flow field in x-d theirection and y-direction induced by the air flow across the gas-water interface. Values were linearly interpolated for points between values from the grid.

Then, a self-written LabVIEW script was used to simulate the Brownian motion of a particle with a chosen diffusion constant starting at a random position in the chamber and propagating along the flow vector field in 10ms time steps. To do this, we look at the random square displacement of a particle with diffusion constant D:

$$x^2 = Dt \tag{14}$$

As this particle can move in two directions, left and right, this becomes

$$x^2 = 2Dt \tag{15}$$

Expanding this into two dimensions, we get:

$$x_{2D}^2 = x_x^2 + x_y^2 \text{ and therefore: } x_{x,y} = \sqrt{x_x^2 + x_y^2} = \sqrt{2Dt + 2Dt} = \sqrt{4Dt} \tag{16}$$

We then inserted the diffusion constant of a 61mer DNA *Mast et al., 2013* of $97.04 \cdot 10^{(-12)} \frac{m^2}{s}$ and a random unit vector phi with values between [–1,1]. $\vec{u}$ and $\vec{v}$ represent the exported laminar flow field data from Comsol:

$$\text{x displacements:} \sqrt{4 \cdot 97.04 \cdot 10^{(-12)} \cdot dt} \cdot phi + dt \cdot \vec{u} \tag{17}$$

$$\text{y displacements:} \sqrt{4 \cdot 97.04 \cdot 10^{(-12)} \cdot dt} \cdot phi + dt \cdot \vec{v} \tag{18}$$

The particle was then displaced according to *Equations 17; 18* with a timestep of 10ms for a total of 35 min. Along its path, the respective local $Mg^{2+}$ and 61mer DNA concentration was plotted, yielding the graph displayed in main text *Figure 3d*. The path was overlaid with the original Comsol simulation graphic of the salt concentration distribution in *Figure 3c*.

## Appendix 8

## Hydrolysis estimation

To estimate the hydrolysis rate of a 24-mer RNA strand deployed in the conditions used in *Figure 3* (45 °C, pH 7, 50–200 µM Mg$^{2+}$, 4–12 mM NaCl), we used a model from literature to derive the hydrolysis rate in dependence of ion concentrations, temperature, and pH (*Li and Breaker, 1999*):

$$k_{hyd}[1/min] = L \cdot k_{bg} \cdot 10^{0.983 \cdot (pH-6)} \cdot 10^{-0.24 \cdot (3.16-[K^+])} \cdot 10^{(0.07 \cdot (T-23))} \cdot 3.57 \cdot [K^+]^{-0.419} \cdot 69.3 \cdot [Mg^{2+}]^{0.8}$$

(19)

Here, $L$ denotes the length of the oligomer (24 in this case), $k_{bg}$ represents a background hydrolysis rate of 1.3E-9 [1 /min], and all concentrations ([Mg$^{2+}$ and [K$^+$]]) are given in Molar. We used Na$^+$ instead of K$^+$ in the experiment, but assumed here, that the K$^+$ correction would be similar for this calculation.

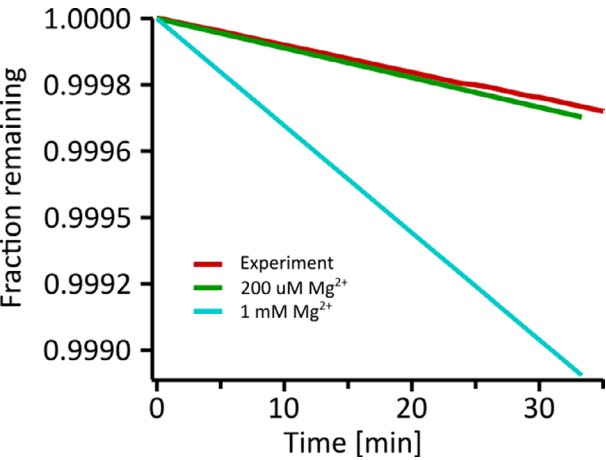

**Appendix 8—figure 1.** Theoretical hydrolysis of RNA in the deployed experimental conditions calculated for the Monte-Carlo trace of Fig. *Figure 3* as well as for constant Mg$^{2+}$ concentrations.

Using the trace of the Monte-Carlo random walk shown in *Figure 3(c)*, we calculated the hydrolysis rate from *Equation 19* for each time step of 10ms iteratively. The resulting relative fraction of remaining oligomers over time is plotted in *Appendix 8—figure 1*. It also shows two comparison graphs, where a constant Mg$^{2+}$ concentration was assumed over the same course of time. In all cases, RNA is very stable. With an average hydrolysis rate of 7.987E-6 1 /min during the experiment, RNA is approximated to have a halftime of 2.09E+03 hr. Even at 1 mM Mg$^{2+}$, the halftime reduces to 516 hr with a rate of 3.231E-5 1 /min. The maximum concentration of Mg$^{2+}$ in the random walk is around 4 X the starting concentration of 50 µM. The static hydrolysis curve at 200 µM Mg$^{2+}$ is also shown in *Appendix 8—figure 1* with a hydrolysis rate of 8.91703E-6 1 /min and a halftime of 1870 hr. After 35 min under the experimental conditions of *Figure 3*, not more than 1 ‰ of the initial RNA would have hydrolysed. Using a per-base copying rate of a ribozyme polymerase *Tupper and Higgs, 2021*; *Horning and Joyce, 2016* of 72 hr$^{-1}$, a 24-mer strand would be copied every 20 min. The timescale of replication using prebiotically plausible machinery therefore out-competes the hydrolysis in the experimental settings used here. Even assuming a base extension rate of 1 hr$^{-1}$ for non-enzymatic replication (*Walton et al., 2019*), one 24-mer strand would get copied once per day, out-competing the estimated hydrolysis.

Overall, the conditions deployed in the experiment are not harsh on RNA. The vortex brings the nucleic acids down to regimes of low salt, further increasing their lifetime repeatedly.

## Appendix 9

### PCR using Taq polymerase

Replication reactions were performed using the AllTaq PCR Core Kit (QIAGEN). Each reaction contained 2.5 U of AllTaq polymerase, 2 X SYBR Green I, 5 nM template, 0.25 µM of each primer, 200 µM of each dNTP and 0.5 X PCR buffer (contains Tris HCl, KCl, $NH_4SO_4$ and $MgCl_2$).

To distinguish the template from the product strand on a PAGE image, we have added an overhang of 5 A's to the 5' end of each primer. The 51mer template will be extended with the product strand then becoming a 61mer. *Figure 4—figure supplement 1* shows a scheme of the replication cycle. Steps A to E represent the phase of replication in which the original 51mer template is extended step by step, first to a 56mer and finally to the 61mer product. Once the template is extended, the reduced replication cycle F to G (*Figure 4—figure supplement 1* green boxes) begins, in which the concentration of template and product strands increases exponentially. Once the initial amount of template (5 nM) is consumed, the reaction can only be represented by the reduced scheme.

To confirm that the reaction in the chamber followed the predicted scheme, we performed a series of control experiments. *Figure 4—figure supplement 2* shows the resulting PAGE gels and the temperature protocol. The experiment was performed either in the microfluidic chamber (*Figure 4—video 1* and *Figure 4—video 4*) or in a test tube inside a thermocycler (*Figure 4—figure supplement 2*). After a heat activation step of 95 °C for Taq polymerase, the temperature was kept constant at 68 °C in the microfluidic chamber, while in the thermocycler we followed the PCR protocol for Taq, in which the sample underwent multiple cycles of replication (*Figure 4—figure supplement 2*). After 95 °C, the primers are given time to anneal by cooling the sample to 52 °C, followed by a replication step at 68 °C, where Taq is at its peak performance. This temperature protocol is then repeated an additional 39 times to complete the 40 cycles, and the sample is then cooled to 4 °C, extracted, and then stored at –20 °C until PAGE analysis.

*Figure 4—figure supplement 2c* shows the PAGE results for the test tube samples in the thermocycler. On the left, the triplicate of the full sample (conditions shown in *Figure 4—figure supplement 2a*) shows primer consumption and the formation of a product band in all cases. To be sure that Taq is not forming an unwanted side product, such as primer dimers, we repeated this experiment without adding the template strand, and indeed no product strand can be detected. Furthermore, the reaction cannot proceed to generate product without having both the reverse and forward primers. In the PAGE gel on the right, experiment triplicates are shown without the forward primer, without the reverse primer, or without primers at all. As expected, no product was formed in any of these cases. Without the addition of DNA (no primers and no template), Taq polymerase does not form a new strand. As a further negative control, we kept a complete sample in the chamber at isothermal 68 °C, analogous to the experiment, and observed no product formation. This is particularly interesting because it shows that without the microfluidic chamber environment, the replication cycle cannot be completed, and the reaction is halted, further emphasizing the need for salt cycling in the chamber experiment.

*Figure 4—figure supplement 2d* shows the PAGE gels of all experiments performed in the microfluidic chamber. While the full sample replicates 2 and 3 show product formation, no product is observed without the addition of the template strand to the reaction mix. When the reaction is run without the reverse primer, forward primer, primers in general, or no DNA at all, no product formation is observed. To show that the replication reaction is only possible when both the gas flow and the water flow are turned on, we repeated the experiment without any fluxes turned on. Here, the full sample was placed in exactly the same microfluidic chamber and kept at isothermal 68 °C as in the other chamber experiments. However, without upconcentration at the interface and without continuous stirring by the gas flow, no product formation can be observed.

Furthermore, we were interested in how many full PCR cycles the sample underwent in the chamber compared to regular PCR using temperature cycling in a test tube. Therefore, we performed the experiment in a test tube with 10 different amounts of temperature cycles as displayed in *Figure 4—figure supplement 2b*. We then compared these results with the samples extracted from the air-flux chamber after 4 hr. *Figure 4—figure supplement 3a* shows the corresponding 15% PAGE image. However, due to losses during the extraction of the sample from the microfluidic chamber, only comparing the gel band intensity of the product from chamber to test tube is not

representative, because the corresponding primer band intensity does not match any of the test tube samples. To account for this, we calculated the ratio of product to primer intensity for all lanes (*Figure 4—figure supplement 3b*). Since losses during extraction are the same for primer as well as for product strands, the ratio of product to primer strands stays unaffected. This reveals that inside the microfluidic chamber, with air and water fluxes turned on, after 4 hr, 10–11 full cycles of replication were performed. Gel intensities were extracted using a self-written LabView tool.

