## [Editor Report · eLife Assessment]

This **important** work shows how a simple geophysical setting of gas flow over a narrow channel of water can create a physical environment that leads to the isothermal replication of nucleic acids. The work presents **compelling** evidence for an isothermal polymerase chain reaction in careful experiments involving evaporation and convective flows, complimented with fluid dynamics simulations. This work will be of interest to scientists working on the origin of life and more broadly, on nucleic acids and diagnostic applications.

---

## [Referee Report · Reviewer #1 (Public review)]

This manuscript from Schwintek and coworkers describes a system in which gas flow across a small channel (10^-4-10^-3 m scale) enables the accumulation of reactants and convective flow. The authors go on to show that this can be used to perform PCR as a model of prebiotic replication.

Strengths:

The manuscript nicely extends the authors' prior work in thermophoresis and convection to gas flows. The demonstration of nucleic acid replication is an exciting one, and an enzyme-catalyzed proof-of-concept is a great first step towards a novel geochemical scenario for prebiotic replication reactions and other prebiotic chemistry.

The manuscript nicely combines theory and experiment, which generally agree well with one another, and it convincingly shows that accumulation can be achieved with gas flows and that it can also be utilized in the same system for what one hopes is a precursor to a model prebiotic reaction. This continues efforts from Braun and Mast over the last 10-15 years extending a phenomenon that was appreciated by physicists and perhaps underappreciated in prebiotic chemistry to increasingly chemically relevant systems and, here, a pilot experiment with a simple biochemical system as a prebiotic model.

I think this is exciting work and will be of broad interest to the prebiotic chemistry community. The techniques described will be useful to the community as well.

Weaknesses:

This work stands well on its own in advancing the field and is well-supported by the evidence presented. The weaknesses below are thus more hopes for future work than limitations of a study that I find to be a complete and well-executed piece of work.

This paper's use of highly evolved protein enzymes is a potential limitation in its direct relevance to prebiotic chemistry. But this is less a limitation of the manuscript than the state of the field after the authors' advances. It will be of interest to see how these systems function in, e.g., RiboPCR (10.1073/pnas.1610103113) and with non enzymatic systems.

Similarly, some of the artifacts in this work (appreciated and noted by the authors) arising from gas bubbles evolving prevent the simulations from fully describing their results. However, gas-liquid interactions were likely important in prebiotic chemistry and the authors note several areas in which these could be important in future systems.

---

## [Referee Report · Reviewer #2 (Public review)]

Schwintek et al. investigated whether a geological setting of a rock pore with water inflow on one end and gas passing over the opening of the pore on the other end could create a non-equilibrium system that sustains nucleic acid reactions under mild conditions. The evaporation of water as the gas passes over it concentrates the solutes at the boundary of evaporation, while the gas flux induces momentum transfer that creates currents in the water that push the concentrated molecules back into the bulk solution. This leads to the creation of steady state regions of differential salt and macromolecule concentrations that can be used to manipulate nucleic acids. First, the authors showed that fluorescent bead behavior in this system closely matched their fluid dynamic simulations. With that validation in hand, the authors next showed that fluorescently-labeled DNA behaved according to their theory as well. Using these insights, the authors performed a FRET experiment that clearly demonstrated hybridization of two DNA strands as they passed through the high Mg++ concentration zone, and, conversely, the dissociation of the strands as they passed through low Mg++ concentration zone. This isothermal hybridization and dissociation of DNA strands allowed the authors to perform an isothermal DNA amplification using a DNA polymerase enzyme. Crucially, the isothermal DNA amplification required the presence of the gas flux and could not be recapitulated using a system that was at equilibrium. These experiments advance our understanding of the geological settings that could support nucleic acid reactions that were key for the origin of life.

The presented data compellingly supports the conclusions made by the authors. In the revised submission, the authors have made convincing arguments supported by simulations that the present findings obtained with DNA would translate to RNA as well, thus making this work highly relevant for the field of origin of life.

A potential future experiment the authors could consider includes performing a prebiotically relevant reaction, such as non-enzymatic primer extension or ligation, in the described model of the rock pore geological setting.

---

## [Author Response]

The following is the authors’ response to the original reviews.

**Public Reviews:**

**Reviewer #1 (Public review):**
This manuscript from Schwintek and coworkers describes a system in which gas flow across a small channel (10^-4-10^-3 m scale) enables the accumulation of reactants and convective flow. The authors go on to show that this can be used to perform PCR as a model of prebiotic replication.Strengths:The manuscript nicely extends the authors' prior work in thermophoresis and convection to gas flows. The demonstration of nucleic acid replication is an exciting one, and an enzyme-catalyzed proof-of-concept is a great first step towards a novel geochemical scenario for prebiotic replication reactions and other prebiotic chemistry.The manuscript nicely combines theory and experiment, which generally agree well with one another, and it convincingly shows that accumulation can be achieved with gas flows and that it can also be utilized in the same system for what one hopes is a precursor to a model prebiotic reaction. This continues efforts from Braun and Mast over the last 10-15 years extending a phenomenon that was appreciated by physicists and perhaps underappreciated in prebiotic chemistry to increasingly chemically relevant systems and, here, a pilot experiment with a simple biochemical system as a prebiotic model.I think this is exciting work and will be of broad interest to the prebiotic chemistry community.Weaknesses:The manuscript states: "The micro scale gas-water evaporation interface consisted of a 1.5 mm wide and 250 µm thick channel that carried an upward pure water flow of 4 nl/s ≈ 10 µm/s perpendicular to an air flow of about 250 ml/min ≈ 10 m/s." This was a bit confusing on first read because Figure 2 appears to show a larger channel - based on the scale bar, it appears to be about 2 mm across on the short axis and 5 mm across on the long axis. From reading the methods, one understands the thickness is associated with the Teflon, but the 1.5 mm dimension is still a bit confusing (and what is the dimension in the long axis?) It is a little hard to tell which portion (perhaps all?) of the image is the channel. This is because discontinuities are present on the left and right sides of the experimental panels (consistent with the image showing material beyond the channel), but not the simulated panels. Based on the authors' description of the apparatus (sapphire/CNC machined Teflon/sapphire) it sounds like the geometry is well-known to them. Clarifying what is going on here (and perhaps supplying the source images for the machined Teflon) would be helpful.

We understand. We will update the figures to better show dimensions of the experimental chamber. We will also add a more complete Figure in the supplementary information. Part of the complexity of the chamber however stems from the fact that the same chamber design has also been used to create defined temperature gradients which are not necessary and thus the chamber is much more complex than necessary.

We added the scheme of the whole PTFE Chip to Figure 2 in the top left corner, indicating the ROI shown in the fluorescence micrographs. Additionally, the channel walls are now clearly indicated by white dotted lines. The dimensions of the setup are now shown clearer, by showing the total width of the channel as well as its height until the gas flux channel, as well as its depth. Changed caption of the figure accordingly and it now reads: “[…] The PTFE chip cutout in the top left corner shows the ROI used for the micrographs. The color scale is equal for both simulation and experiment and Channel dimensions are 4 x 1.5 x 0.25 mm as indicated. Dotted lines visualize the location of the channel walls. […]“

The data shown in Figure 2d nicely shows nonrandom residuals (for experimental values vs. simulated) that are most pronounced at t~12 m and t~40-60m. It seems like this is (1) because some symmetry-breaking occurs that isn't accounted for by the model, and perhaps (2) because of the fact that these data are n=1. I think discussing what's going on with (1) would greatly improve the paper, and performing additional replicates to address (2) would be very informative and enhance the paper. Perhaps the negative and positive residuals would change sign in some, but not all, additional replicates?

To address this, we will show two more replicates of the experiment and include them in Figure 2.

We are seeing two effects when we compare fluorescence measurements of the experiments.

Firstly, degassing of water causes the formation of air-bubbles, which are then transported upwards to the interface, disrupting fluorescence measurements. This, however, mostly occurs in experiments with elevated temperatures for PCR reactions, such as displayed in Figure 4.

Secondly, due to the high surface tension of water, the interface is quite flexible. As the inflow and evaporation work to balance each other, the shape of the interface adjusts, leading to alterations in the circular flow fields below.

Thus the conditions, while overall being in steady state, show some fluctuations. The strong dependence on interface shape is also seen in the simulation. However, modeling a dynamic interface shape is not so easy to accomplish, so we had to stick to one geometry setting. Again here, the added movies of two more experiments should clarify this issue.

We performed three more replicates of the experiment and included the averaged data points together with their respective standard deviation as error bars in Figure 2d. Additionally, the videos of each individual repeat are now added to the supplementary files for the reader to better understand where the strong fluctuations around half an hour come from. The Figure caption was adjusted to “ […] The maximum relative concentration of DNA increased within an hour to ~30 X the initial concentration, with the trend following the simulation. Error bars are the standard deviation from four independent measurements. […].

The main text was also changed to better explain how the fluctuations impact the measurements: […] Water continuously evaporated at the interface, but nucleic acids remained in the aqueous phase accumulating near the interface. They could only escape downward either by diffusion or by the vortex induced by the gas flowing across the interface, pushing the molecules back deeper into the bulk (See the flow lines in Fig2(b) taken from the simulation). As the gas flow continuously removed excess vapor, the evaporation rate remained constant. Thus, except for fluctuations, a stable interface shape should be expected. However, due to the high surface tension of water, the interface is very flexible. As the inflow and evaporation work to balance each other, the shape of the interface adjusts, likely in response to small fluctuations in gas pressure and spatial variations in water surface tension. This is leading to alterations in the circular flow fields below (Supplementary Movie 2).

As these fluctuations are difficult to simulate, we decided to stick with one interface shape, matching evaporation and inflow speeds. The evaporation rate at the interface was therefore set to be proportional to the vapor concentration gradient and varied spatially along the interface between 5 and 10.5 µm/s (See Suppl. Fig. VI.1(d)). Using the known diffusion coefficient of 95 µm²/s for the 63mer[9]}, the simulation closely matched the experimental results. In both cases, DNA accumulated in regions with circular flow patterns driven by the gas flux (Fig.2(b), right panel).

5 minutes after starting the experiment, the maximum DNA accumulation was 3-fold, while after one hour of evaporation, around 30-fold accumulation was observed. Due to molecules residing in very shallow volumes when directly at the interface, the fluorescence signal can vary drastically compared to measurements deeper in the bulk. This can be seen in the fluctuations between independent measurements (See Supplementary Movies 2b,2b,2c), especially around 0.5~h shown in Figure 2(d). The simulated maximum accumulation followed the experimental results and starts saturating after about one hour (Fig.2(d)). […]”

The authors will most likely be familiar with the work of Victor Ugaz and colleagues, in which they demonstrated Rayleigh-Bénard-driven PCR in convection cells (10.1126/science.298.5594.793, 10.1002/anie.200700306). Not including some discussion of this work is an unfortunate oversight, and addressing it would significantly improve the manuscript and provide some valuable context to readers. Something of particular interest would be their observation that wide circular cells gave chaotic temperature profiles relative to narrow ones and that these improved PCR amplification (10.1002/anie.201004217). I think contextualizing the results shown here in light of this paper would be helpful.

Thanks for pointing this out and reminding us. We apologize. We agree that the chaotic trajectories within Rayleigh-Bénard convection cells lead to temperature oscillations similar to the salt variations in our gas-flux system. Although the convection-driven PCR in Rayleigh-Bénard is not isothermal like our system, it provides a useful point of comparison and context for understanding environments that can support full replication cycles. We will add a section comparing approaches and giving some comparison into the history of convective PCR and how these relate to the new isothermal implementation.

We added a main text paragraph after the last paragraph in section “Strand Separation Dynamics”: “[…]Rayleigh-Bénard convection cells generate similar patterns to those seen in Fig. 3(c) The oscillations in salt concentration resemble the temperature fluctuations observed in convection-based PCR reactions from earlier studies [32,33], which showed that chaotic temperature variations, compared to periodic ones, enhanced the efficiency of the PCR reaction.[…]

Again, it appears n=1 is shown for Figure 4a-c - the source of the title claim of the paper - and showing some replicates and perhaps discussing them in the context of prior work would enhance the manuscript.

We appreciate the reviewer for bringing this to our attention. We will now include the two additional repeats for the data shown in Figure 4c, while the repeats of the PAGE measurements are already displayed in Supplementary Fig. IX.2. Initially, we chose not to show the repeats in Figure 4c due to the dynamic and variable nature of the system. These variations are primarily caused by differences at the water-air interface, attributed to the high surface tension of water. Additionally, the stochastic formation of air bubbles in the inflow—despite our best efforts to avoid them—led to fluctuations in the fluorescence measurements across experiments. These bubbles cause a significant drop in fluorescence in a region of interest (ROI) until the area is refilled with the sample.

Unlike our RNA-focused experiments, PCR requires high temperatures and degassing a PCR master mix effectively is challenging in this context. While we believe our chamber design is sufficiently gas-tight to prevent air from diffusing in, the high surface-to-volume ratio in microfluidics makes degassing highly effective, particularly at elevated temperatures. We anticipate that switching to RNA experiments at lower temperatures will mitigate this issue, which is also relevant in a prebiotic context.

The reviewer’s comments are valid and prompt us to fully display these aspects of the system. We will now include these repeats in Figure 4c to give readers a deeper understanding of the experiment's dynamics. Additionally, we will provide videos of all three repeats, allowing readers to better grasp the nature of the fluctuations in SYBR Green fluorescence depicted in Figure 4c.

The data from the triplicates are now added to Figure 4c, showing how air bubbles, forming through degassing at the high temperatures required for Taq polymerase, disrupt the measurement, as they momentarily dry off the channel and stop the reaction until the channel fills again. Figure caption has been adapted and now reads: “[…] Dotted lines show the data from independent repeats. Air bubbles formed through degassing can momentarily disrupt the reaction. […]”

We additionally changed the main text to explain the reader the experimental difficulties: “[…] In other repetitions of the reaction, this increase was sometimes even observed earlier, around the one-hour mark (dotted lines). However, air bubbles nucleated by degassing events rise and temporarily dry out the channel, interrupting the reaction until the liquid refills the channel (Supplementary Movies 4,4b,4c\&5). Despite our best efforts, we were unable to fully prevent this, especially given the high temperatures required for Taq polymerase activity. In an identical setting when the gas- and water flux were switched off, no fluorescence increase was found (See Fig. 4(c) red lines). Fluorescence variations are additionally caused by fluctuations in the position of the gas-water interface, as discussed earlier. […]”

I think some caution is warranted in interpreting the PCR results because a primer-dimer would be of essentially the same length as the product. It appears as though the experiment has worked as described, but it's very difficult to be certain of this given this limitation. Doing the PCR with a significantly longer amplicon would be ideal, or alternately discussing this possible limitation would be helpful to the readers in managing expectations.

This is a good point and should be discussed more in the manuscript. Our gel electrophoresis is capable of distinguishing between replicate and primer dimers. We know this since we were optimizing the primers and template sequences to minimize primer dimers, making it distinguishable from the desired 61mer product. That said, all of the experiments performed without a template strand added did not show any band in the vicinity of the product band after 4h of reaction, in contrast to the experiments with template, presenting a strong argument against the presence of primer dimers.

We added a main text section explaining this to the reader: “[…]Suppl. Fig. IX.2 shows all independent repeats of the corresponding experiments. No product was detected in any of these cases, ruling out reaction limitations such as primer dimer formation. Primer dimers would form even in the absence of a template strand and would be identifiable through gel electrophoresis. As Taq polymerase requires a significant overlap between the two dimers to bind, this would result in a shorter product compared to the 61mer used here. […]”

**Reviewer #2 (Public review):**
Schwintek et al. investigated whether a geological setting of a rock pore with water inflow on one end and gas passing over the opening of the pore on the other end could create a non-equilibrium system that sustains nucleic acid reactions under mild conditions. The evaporation of water as the gas passes over it concentrates the solutes at the boundary of evaporation, while the gas flux induces momentum transfer that creates currents in the water that push the concentrated molecules back into the bulk solution. This leads to the creation of steady-state regions of differential salt and macromolecule concentrations that can be used to manipulate nucleic acids. First, the authors showed that fluorescent bead behavior in this system closely matched their fluid dynamic simulations. With that validation in hand, the authors next showed that fluorescently labeled DNA behaved according to their theory as well. Using these insights, the authors performed a FRET experiment that clearly demonstrated the hybridization of two DNA strands as they passed through the high Mg++ concentration zone, and, conversely, the dissociation of the strands as they passed through the low Mg++ concentration zone. This isothermal hybridization and dissociation of DNA strands allowed the authors to perform an isothermal DNA amplification using a DNA polymerase enzyme. Crucially, the isothermal DNA amplification required the presence of the gas flux and could not be recapitulated using a system that was at equilibrium. These experiments advance our understanding of the geological settings that could support nucleic acid reactions that were key to the origin of life.The presented data compellingly supports the conclusions made by the authors. To increase the relevance of the work for the origin of life field, the following experiments are suggested:(1) While the central premise of this work is that RNA degradation presents a risk for strand separation strategies relying on elevated temperatures, all of the work is performed using DNA as the nucleic acid model. I understand the convenience of using DNA, especially in the latter replication experiment, but I think that at least the FRET experiments could be performed using RNA instead of DNA.

We understand the request only partially. The modification brought about by the two dye molecules in the FRET probe to be able to probe salt concentrations by melting is of course much larger than the change of the backbone from RNA to DNA. This was the reason why we rather used the much more stable DNA construct which is also manufactured at a lower cost and in much higher purity also with the modifications. But we think the melting temperature characteristics of RNA and DNA in this range is enough known that we can use DNA instead of RNA for probing the salt concentration in our flow cycling.

Only at extreme conditions of pH and salt, RNA degradation through transesterification, especially under alkaline conditions is at least several orders of magnitude faster than spontaneous degradative mechanisms acting upon DNA [Li, Y., & Breaker, R. R. (1999). Kinetics of RNA degradation by specific base catalysis of transesterification involving the 2 ‘-hydroxyl group. Journal of the American Chemical Society, 121(23), 5364-5372.]. The work presented in this article is however focussed on hybridization dynamics of nucleic acids. Here, RNA and DNA share similar properties regarding the formation of double strands and their respective melting temperatures. While RNA has been shown to form more stable duplex structures exhibiting higher melting temperatures compared to DNA [Dimitrov, R. A., & Zuker, M. (2004). Prediction of hybridization and melting for double-stranded nucleic acids. Biophysical Journal, 87(1), 215-226.], the general impact of changes in salt, temperature and pH [Mariani, A., Bonfio, C., Johnson, C. M., & Sutherland, J. D. (2018). pH-Driven RNA strand separation under prebiotically plausible conditions. Biochemistry, 57(45), 6382-6386.] on respective melting temperatures follows the same trend for both nucleic acid types. Also the diffusive properties of RNA and DNA are very similar [Baaske, P., Weinert, F. M., Duhr, S., Lemke, K. H., Russell, M. J., & Braun, D. (2007). Extreme accumulation of nucleotides in simulated hydrothermal pore systems. Proceedings of the National Academy of Sciences, 104(22), 9346-9351.].

Since this work is a proof of principle for the discussed environment being able to host nucleic acid replication, we aimed to avoid second order effects such as degradation by hydrolysis by using DNA as a proxy polymer. This enabled us to focus on the physical effects of the environment on local salt and nucleic acid concentration. The experiments performed with FRET are used to visualize local salt concentration changes and their impact on the melting temperature of dissolved nucleic acids. While performing these experiments with RNA would without doubt cover a broader application within the field of origin of life, we aimed at a step-by-step / proof of principle approach, especially since the environmental phenomena studied here have not been previously investigated in the OOL context. Incorporating RNA-related complexity into this system should however be addressed in future studies. This will likely require modifications to the experimental boundary conditions, such as adjusting pH, temperature, and salt concentration, to account for the greater duplex stability of RNA. For instance, lowering the pH would reduce the RNA melting temperature [Ianeselli, A., Atienza, M., Kudella, P. W., Gerland, U., Mast, C. B., & Braun, D. (2022). Water cycles in a Hadean CO2 atmosphere drive the evolution of long DNA. Nature Physics, 18(5), 579-585.].

(2) Additionally, showing that RNA does not degrade under the conditions employed by the authors (I am particularly worried about the high Mg++ zones created by the flux) would further strengthen the already very strong and compelling work.

Based on literature values for hydrolysis rates of RNA [Li, Y., & Breaker, R. R. (1999). Kinetics of RNA degradation by specific base catalysis of transesterification involving the 2 ‘-hydroxyl group. Journal of the American Chemical Society, 121(23), 5364-5372.], we estimate RNA to have a half-life of multiple months under the deployed conditions in the FRET experiment (High concentration zones contain <1mM of Mg2+). Additionally, dsRNA is multiple orders of magnitude more stable than ssRNA with regards to degradation through hydrolysis [Zhang, K., Hodge, J., Chatterjee, A., Moon, T. S., & Parker, K. M. (2021). Duplex structure of double-stranded RNA provides stability against hydrolysis relative to single-stranded RNA. Environmental Science & Technology, 55(12), 8045-8053.], improving RNA stability especially in zones of high FRET signal. Furthermore, at the neutral pH deployed in this work, RNA does not readily degrade. In previous work from our lab [Salditt, A., Karr, L., Salibi, E., Le Vay, K., Braun, D., & Mutschler, H. (2023). Ribozyme-mediated RNA synthesis and replication in a model Hadean microenvironment. Nature Communications, 14(1), 1495.], we showed that the lifetime of RNA under conditions reaching 40mM Mg2+ at the air-water interface at 45°C was sufficient to support ribozymatically mediated ligation reactions in experiments lasting multiple hours.

With that in mind, gaining insight into the median Mg2+ concentration across multiple averaged nucleic acid trajectories in our system (see Fig. 3c&d) and numerically convoluting this with hydrolysis dynamics from literature would be highly valuable. We anticipate that longer residence times in trajectories distant from the interface will improve RNA stability compared to a system with uniformly high Mg2+ concentrations.

Added a new Supplementary section for this. We used the trace from Figure 3(c) and calculated the hydrolysis rate for each timestep by using literature values from RNA [Li, Y., & Breaker, R. R. (1999). Kinetics of RNA degradation by specific base catalysis of transesterification involving the 2 ‘-hydroxyl group. Journal of the American Chemical Society, 121(23), 5364-5372.]. We conclude that the conditions deployed for the experiment are not harsh on RNA, with hydrolysis rates in the E-6 1/min regime. The figure below (also now in the supplementary information) shows the hydrolysis of RNA deployed under the conditions of the experiment in Figure 3. RNA is not expected to hydrolyze under these conditions and timescales, in which a replication reaction would occur. With a half life of around 83 days, even a prebiotically plausible – very slow – replication reaction would not be constrained by hydrolysis boundary conditions in this scenario.

Referenced to this section in the supplementary information in the maintext: […] In the experimental conditions used here, RNA would also not readily degrade, even if the strand enters the high salt regimes (See Suppl. Sec. IX). Using literature values for hydrolysis rates under the deployed conditions, we estimate dissolved RNA to have a half life of around 83 days. […]

(3) Finally, I am curious whether the authors have considered designing a simulation or experiment that uses the imidazole- or 2′,3′-cyclic phosphate-activated ribonucleotides. For instance, a fully paired RNA duplex and a fluorescently-labeled primer could be incubated in the presence of activated ribonucleotides +/- flux and subsequently analyzed by gel electrophoresis to determine how much primer extension has occurred. The reason for this suggestion is that, due to the slow kinetics of chemical primer extension, the reannealing of the fully complementary strands as they pass through the high Mg++ zone, which is required for primer extension, may outcompete the primer extension reaction. In the case of the DNA polymerase, the enzymatic catalysis likely outcompetes the reannealing, but this may not recapitulate the uncatalyzed chemical reaction.

This is certainly on our to-do list for future experiments in this setting. Our current focus is on templated ligation rather than templated polymerization and we are working hard to implement RNA-only enzyme-free ligation chain reaction, based on more optimized parameters for the templated ligation from 2’3’-cyclic phosphate activation that was just published [High-Fidelity RNA Copying via 2′,3′-Cyclic Phosphate Ligation, Adriana C. Serrão, Sreekar Wunnava, Avinash V. Dass, Lennard Ufer, Philipp Schwintek, Christof B. Mast, and Dieter Braun, JACS doi.org/10.1021/jacs.3c10813 (2024)]. But we first would try this at an air-water interface which was shown to work with RNA in a temperature gradient [Ribozyme-mediated RNA synthesis and replication in a model Hadean microenvironment, Annalena Salditt, Leonie Karr, Elia Salibi, Kristian Le Vay, Dieter Braun & Hannes Mutschler, Nature Communications doi.org/10.1038/s41467-023-37206-4 (2023)] before making the jump to the isothermal setting we describe here. So we can understand the question, but it was good practice also in the past to first get to know the setting with PCR, then jump to RNA.

**Recommendations for the authors:**

**Reviewer #2 (Recommendations for the authors):**
(1) Could the authors comment on the likelihood of the geological environments where the water inflow velocity equals the evaporation velocity?

This is an important point to mention in the manuscript, thank you for pointing that out. To produce a defined experiment, we were pushing the water out with a syringe pump, but regulated in a way that the evaporation was matching our flow rate. We imagine that a real system will self-regulate the inflow of the water column on the one hand side by a more complex geometry of the gas flow, matching the evaporation with the reflow of water automatically. The interface would either recede or move closer to the gas flux, depending on whether the inflow exceeds or falls short of the evaporation rate. As the interface moves closer, evaporation speeds up, while moving away slows it down. This dynamic process stabilizes the system, with surface tension ultimately fixing the interface in place.

We have seen a bit of this dynamic already in the experiments, could however so far not yet find a good geometry within our 2-dimensional constant thickness geometry to make it work for a longer time. Very likely having a 3-dimensional reservoir of water with less frictional forces would be able to do this, but this would require a full redesign of a multi-thickness microfluidics. The more we think about it, the more we envisage to make the next implementation of the experiment with a real porous volcanic rock inside a humidity chamber that simulates a full 6h prebiotic day. But then we would lose the whole reproducibility of the experiment, but likely gain a way that recondensation of water by dew in a cold morning is refilling the water reservoirs in the rocks again. Sorry that I am regressing towards experiments in the future.

We added a paragraph after the second paragraph in Results and Discussion.

It now reads: […] For a real early Earth environment we envision a system that self-regulates the water column's inflow by automatically balancing evaporation with capillary flows. The interface adjusts its position relative to the gas flux, moving closer if the inflow is less than the evaporation rate, or receding if it exceeds it. When the interface nears the gas flux, evaporation accelerates, while moving it away slows evaporation. This dynamic process stabilizes the system, with surface tension ultimately fixing the interface's position. […]

(2) Could the authors speculate on using gases other than ambient air to provide the flux and possibly even chemical energy? For example, using carbonyl sulfide or vaporized methyl isocyanide could drive amino acid and nucleotide activation, respectively, at the gas-water interface.

This is an interesting prospect for future work with this system. We thought also about introducing ammonia for pH control and possible reactions. We were amazed in the past that having CO2 instead of air had a profound impact on the replication and the strand separation [Water cycles in a Hadean CO2 atmosphere drive the evolution of long DNA, Alan Ianeselli, Miguel Atienza, Patrick Kudella, Ulrich Gerland, Christof Mast & Dieter Braun, Nature Physics doi.org/10.1038/s41567-022-01516-z (2022)]. So going more in this direction absolutely makes sense and as it acts mostly on the length-selectively accumulated molecules at the interface, only the selected molecules will be affected, which adds to the selection pressure of early evolutionary scenarios.

Of course, in the manuscript, we use ambient air as a proxy for any gas, focusing primarily on the energy introduced through momentum transfer and evaporation. We speculate that soluble gasses could establish chemical gradients, such as pH or redox potential, from the bulk solution to the interface, similar to the Mg2+ accumulation shown in Figure 3c. The nature of these gradients would depend on each gas's solubility and diffusivity. We have already observed such effects in thermal gradients [Keil, L. M., Möller, F. M., Kieß, M., Kudella, P. W., & Mast, C. B. (2017). Proton gradients and pH oscillations emerge from heat flow at the microscale. Nature communications, 8(1), 1897.] and finding similar behavior in an isothermal environment would be a significant discovery.

Added a paragraph in the Conclusion to showcase this: […] Furthermore we expect that other gases, such as CO2, could establish chemical gradients in this environment. Such gradients have been observed in thermal gradients before [23] and finding similar behaviour in an isothermal environment would be a significant discovery.[…]

(3) Line 162: Instead of "risk," I suggest using "rate".

Thanks for pointing this out! Will be changed.

Fixed.

(4) Using FRET of a DNA duplex as an indicator of salt concentration is a decent proxy, but a more direct measurement of salt concentration would provide further merit to the explicit statement that it is the salt concentration that is changing in the system and not another hidden parameter.

Directly observing salt concentration using microscopy is a difficult task. While there are dyes that change their fluorescence depending on the local Na+ or Mg2+ concentration, they are not operating differentially, i.e. by making a ratio between two color channels. Only then we are not running into artifacts from the dye molecules being accumulated by the non-equilibrium settings. We were able to do this for pH in the past, but did not find comparable optical salt sensors. This is the reason we ended up with a FRET pair, with the advantage that we actually probe the strand separation that we are interested in anyhow. Using such a dye in future work would however without a doubt enhance the understanding of not only this system, but also our thermal gradient environments.

(5) Figure 3a: Could the authors add information on "Dried DNA" to the caption? I am assuming this is the DNA that dried off on the sides of the vessel but cannot be sure.

Thanks to the reviewer for pointing this out. This is correct and we will describe this better in the revised manuscript.

Added a sentence in the caption to address this: […] Fluctuations in interface position can dry and redissolve DNA repeatedly (see “Dried DNA” in right panel). […]

(6) Figure 4b and c: How reproducible is this data? Have the authors performed this reaction multiple independent times? If so, this data should be added to the manuscript.

The data from the gel electrophoresis was performed in triplicates and is shown in full in supplementary information. The data in c is hard to reproduce, as the interface is not static and thus ROI measurements are difficult to perform as an average of repeats. Including the data from the independent repeats will however give the reader insight into some of the experimental difficulties, such as air bubbles, which form from degassing as the liquid heats up, that travel upwards to the interface, disrupting the ongoing fluorescence measurements.

This was also pointed out by reviewer 1 and addressed there.

(7) Line 256: "shielding from harmful UV" statement only applies to RNA oligomers as UV light may actually be beneficial for earlier steps during ribonucleoside synthesis. I suggest rephrasing to "shielding nucleic acid oligomers from UV damage.".

Will be adjusted as mentioned.

Fixed.

(8) The final paragraph in the Results and Discussion section would flow better if placed in the Conclusion section.

This is a good point and we will merge results and discussion closer together.

Fixed.

(9) Line 262, "...of early Life" is slightly overstating the conclusions of the study. I suggest rephrasing to "...of nucleic acids that could have supported early life."

This is a fair comment. We thank the reviewer for his detailed analysis of the manuscript!

Changed the phrase to: […]In this work we investigated a prebiotically plausible and abundant geological environment to support the replication of nucleic acids. […]

(10) In references, some of the journal names are in sentence case while others are in title case (see references 23 and 26 for example).

Thanks - this will be fixed.

Fixed.